# ENVIRONMENT-ADVERSARIAL SUB-TASK CURRICULUM FOR EFFICIENT REINFORCEMENT LEARNING

## ABSTRACT

Reinforcement learning (RL)'s efficiency can drastically degrade on long-horizon tasks due to sparse rewards and the learned policy can be fragile to small changes in deployed environments. We address these two challenges by automatically generating a curriculum of tasks with coupled environments. To this end, we train two curriculum policies together with RL: (1) a co-operative planning policy recursively decomposing a hard task into coarse-to-fine sub-task sequences as a tree; and (2) an adversarial policy modifying the environment (e.g., position/size of obstacles) in each sub-task. They are complementary in acquiring more informative feedback for RL: the planning policy provides dense reward of finishing easier sub-tasks while the environment policy modifies these sub-tasks to be adequately challenging and diverse so the RL agent can quickly adapt to different tasks/environments. On the other hand, they are trained using the RL agent's dense feedback on sub-tasks so the sub-task curriculum keeps adaptive to the agent's progress via this "iterative mutual-boosting" scheme. Moreover, the sub-task tree naturally enables an easy-to-hard curriculum for every policy: its top-down construction gradually increases sub-tasks the planning policy needs to generate, while the adversarial training between the environment policy and the RL policy follows a bottom-up traversal that starts from a dense sequence of easier sub-tasks allowing more frequent modifications to the environment. Therefore, jointly training the three policies leads to efficient RL guided by a curriculum progressively improving the sparse reward and generalization. We compare our method with popular RL/planning approaches targeting similar problems and the ones with environment generators or adversarial agents. Thorough experiments on diverse benchmark tasks demonstrate significant advantages of our method on improving RL's efficiency and generalization.

## 1 INTRODUCTION

Although RL achieves breakthrough success or even outperform humans on a few challenging tasks (Lillicrap et al., 2016; Mnih et al., 2015; Florensa et al.), it is highly inefficient when targeting long-horizon tasks due to the sparse rewards collected via interactions. In addition, a policy trained in a specific complicated/simulated environment can be sensitive to small changes in the deployed environment and thus generalizes poorly in practice. Hence, selecting or generating more informative tasks and environments interacting with an agent is an essential challenge on the path towards more efficient, robust, and versatile RL. In this paper, we mainly study goal-conditioned RL (Kaelbling, 1993) whose policy is trained to adapt to any given goal/task: it is challenging, but the resulted policy can be applied to multiple goals/tasks.

The sparse reward problem has motivated different reward shaping/relabeling methods and curiosity-driven exploration for RL, aiming to modify the task reward to be denser than states and actions. Hierarchical RL/planning (Elbanhawi & Simic, 2014; Nasiriany et al., 2019; Jurgenson et al., 2020; Pertsch et al., 2020) decomposes a complicated task/motion by searching a root-to-leaf path of sub-tasks on a tree such that a higher-level task invokes lower-level ones as its actions. However, building the tree requires hierarchical partition of the whole state-action space and prior knowledge to define the sub-tasks, which can be expensive or unavailable. Moreover, the pre-defined sub-tasks can be either too easy or too challenging for the RL-agent in-training. Reward shaping (Laud, 2004) relies on heuristics or external guidance to augment the sparse task-reward with dense rewards for exploring uncertain actions, rarely-visited states, or intermediate goals. Hindsight experience replay (Andrychowicz et al., 2017) and its variants relabels the achieved states as pseudo-goals with nonzero rewards, but many of them might be redundant to provide informative feedback efficiently improving the policy. Hence, automatically modifying the sparse reward or generating sub-tasks to

provide the most informative feedback maximizing the RL agent's learning progress is still an open problem, and resolving it can result in more efficient interactive learning.

Another primary challenge in RL is how to improve its robustness and generalization to small changes in the environments. A growing number of recent studies (Pinto et al., 2017; Vinitsky et al., 2020; Ferguson & Law, 2018) show that RL policy can be vulnerable to small perturbations in its inputs so training an adversarial agent to perturb the inputs or compete with the original agent may improve RL's robustness. This paper mainly focuses on improving RL policy's tolerance to the physical differences between the training environment and possible deployed environment, e.g., changes of the position and size of obstacles/objects. In practice, this is important for deploying an RL policy trained in a simulator successfully to a realistic environment. Moreover, when addressing the sparse reward problem, the engineered sub-tasks or relabeled goals might be redundant or too easy to provide effective feedback to RL. However, adversarially modifying their environments can effectively improve their utility and RL policy's robust generalization against diverse perturbed environments. Although modifying environment models has been recently studied to assist RL (Co-Reyes et al., 2020; Gur et al., 2021; Wang et al., 2019), directly applying this strategy to long-horizon tasks might make them even more challenging and reward-sparse, hence detrimental to RL's efficiency.

In this paper, we address the above two challenges by automatically generating a curriculum of sub-tasks with adversarially perturbed environments to train the RL agent, where the curriculum is adaptive to the agent's learning progress. To this end, we propose a marriage of sub-tasking and adversarial environments. The former decomposes a challenging long-horizon task into easier sub-tasks offering dense rewards, and the latter modifies each sub-task to be sufficiently challenging yet effective for improving RL policy's tolerance to perturbed environments. As illustrated in Fig. 1, our approach, "environment-adversarial sub-Task curriculum (EAT-C)", generates a tree-structured curriculum by (1) a path-planning policy that recursively decomposes an assigned task (e.g., a state-goal pair $(s, g)$) to sequences of sub-tasks (e.g., navigating between

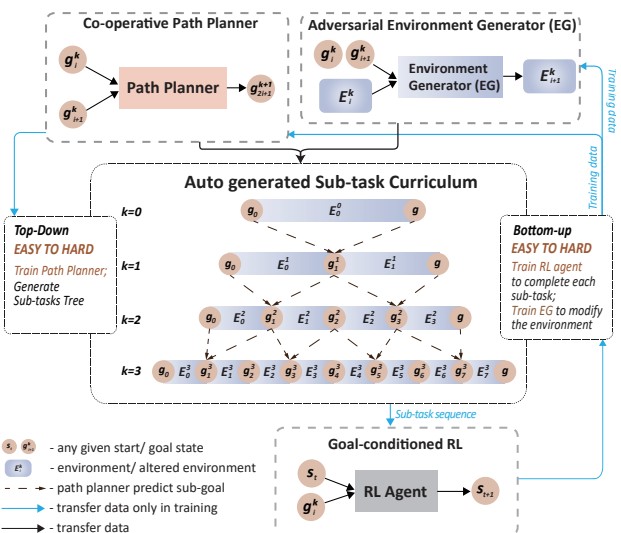

Figure 1: **Main structure of EAT-C:** The path-planner recursively generates a sub-task tree for a given task $(g_0, g)$, while the environment generator (EG) adversarially modifies the environment of each sub-task. RL agent is trained on a bottom-up curriculum and its collected data are used to train the path-planner and EG.

consecutive sub-goals) with varying granularity ($k = 0, \ldots, 3$); and (2) an environment policy that adversarially modifies each sub-task's environment. The training objective for the path-planner is finding the most cost-efficient/shortest path in each level $k$, while the objective for the environment policy is to minimize the expected return of the RL agent on all sub-tasks. In EAT-C, training these two policies does not require external supervision or prior knowledge, we instead collect the time costs and rewards of the RL agent on previous sub-tasks to train them towards generating better curricula adaptive to RL progress, which then guide the training of the next episode.

Together with the RL policy, the two curriculum policies can efficiently learn from each other by iterating the above mutual-boosting scheme on the tree-structured curriculum of sub-tasks. A key advantage of the tree is that it naturally enables an easy-to-hard curriculum to train each of the three policies: (1) the top-down construction of the tree trains the path-planner to merely interpolate a few sub-goals between $(s, g)$ at first and then gradually increases their number for more fine-grained and challenging planning; (2) the adversarial training between the environment policy and the RL policy follows a bottom-up traversal of the tree, i.e., they start from learning many easier sub-tasks and more frequent perturbations between $(s, g)$ respectively, which are easy for both policies, and then progressively evolve to handle more challenging cases. i.e., a few long-horizon sub-tasks allow less-frequent environment changes between $(s, g)$. In experiments for discrete navigation and continuous

control tasks, we show that EAT-C is considerably efficient in learning to generalize to perturbed environments during finishing long-horizon tasks. Compared with recent RL methods equipped with hand-crafted curricula, environment generators, hierarchical policies, or path-planning, EAT-C exhibits promising advantages on efficiency and generalization to different tasks and environments.

## 2 RELATED WORK

Goal-conditioned RL (Pong et al., 2018; Kaelbling, 1993) takes a goal as an additional input to its policy and aims to handle different goals/tasks by the same policy. However, it requires extensive exploration and costly training on various goals but it still easily fails to reach distant goals in practice. Goal relabeling and reward shaping (Andrychowicz et al., 2017; Nasiriany et al., 2019) have been studied to mitigate these issues by modifying the task rewards to be dense but they introduce extra data bias and cannot control the utility of modified goals/rewards to the targeted ones. In order to provide tasks at the appropriate level of difficulty for the agent to learn, Held et al. (2018) and Racanière et al. (2019) train a goal/task generator based on the agent's performance on existing tasks. But it is usually non-trivial to determine and tune the appropriate difficulty level for each training stage. In EAT-C, an RL agent achieves dense rewards by finishing a sequence of easier sub-goals towards the final goal, while the adversarial EG modifies each sub-task to be sufficiently challenging and diverse from other sub-tasks. Such sub-task curricula lead to more efficient goal-conditioned RL. Moreover, we do not need to carefully tune the difficulty levels of sub-tasks because the path-planner and EG together automatically generate the most informative tasks for each training stage of RL.

Planning algorithms (Sutton & Barto, 2018; LaValle, 2006) are helpful to solving long-horizon tasks (Levine et al., 2011) by interpolating intermediate sub-goals. In navigation, planning usually refers to finding the shortest path between two nodes(states) on a graph (LaValle, 2006; Dayan & Hinton, 1993) but a distance metric more accurate than Euclidean distance is usually unavailable. Moreover, it suffers from local minimal and performs poorly in narrow regions (Koren & Borenstein, 1991). Sequentially planning (Schmidhuber & Wahnsiedler, 1993; Zhang et al., 2020) sub-goals from the starting state to the goal state is inefficient in complex tasks, as it needs to search in a large space. In RL, planning requires an environment model or learns a value function to improve the policy (Levine et al., 2011; Lau & Kuffner, 2005; Elbanhawi & Simic, 2014), which can be as challenging as model-free RL. Control-based methods require accurate models for both the robot and the environment (Howard & Kelly, 2007; Werling et al., 2012), as well as an accurate distance metric (Eysenbach et al., 2019), which can be a rather daunting task. In EAT-C, we train a path-planner that "learns to plan" and to directly generate the shortest (in terms of time cost) path for the RL agent. Hence, it requires neither expensive search procedures nor a predefined distance metric. Moreover, it does not rely on learning a world/robot model or a value function. The easy-to-hard curriculum and dense feedback from the RL agent make it efficient to train. Furthermore, it does not need to plan for every step but only a few crucial waypoints.

Several recent works (Wang et al., 2019; Portelas et al., 2019; Gur et al., 2021) show that the efficiency and generalization of RL can be improved by generating/selecting different environments for training. Wang et al. (2019) studies a method "POET" to pair an adversarial environment generator with two agents to improve the generalization of the RL policy to novel environments. Portelas et al. (2019) proposes a teacher algorithm to sample environments based on a Gaussian mixture model capturing both the diversity of environments and the learning progress on them. This leads to a curriculum improving the efficiency of RL. However, the generated adversarial environments can be infeasible or too challenging for the RL agent and it is non-trivial to control their hardness. Moreover, modeling the distribution of environments requires collecting many samples from them and evaluating the RL policy on them, which can be costly and inaccurate especially for complicated environments. In EAT-C, we adversarially modify the environment for each easier sub-task, whose hardness is controlled by both the path planner and EG, so the modified sub-task is still feasible for the RL agent to learn. Moreover, a policy network for EG facilitates the modification of complicated environments and training it is efficient in our mutual boosting scheme with an easy-to-hard curriculum.

Hierarchical planning (HP) methods (Nasiriany et al., 2019; Jurgenson et al., 2020; Pertsch et al., 2020) search for a sequence of sub-goal on a tree to guide the exploration of an agent but building the hierarchical partition of all possible sub-goals can be expensive. A conventional hierarchical planning algorithm (Kaelbling & Lozano-Pérez, 2011) also learns to predict sub-tasks of a tree structure based

on a set of pre-defined motion primitives. Hierarchical RL (HRL) for goal-reaching tasks has been recently studied in (Zhang et al., 2021; Nachum et al., 2018). Shu et al. (2018) trains the RL agent on a human-designed curriculum of tree-structured sub-goals. They learn a sequence of primeval policies towards finishing complicated tasks, where the higher-level policies decompose a complex task into easier sub-tasks or motion primitives that can be addressed by the lower-level policies or controllers. However, the hierarchy of sub-tasks need to be either determined by prior knowledge or discovered automatically, which is usually challenging due to the huge space of possible sub-tasks. In contrast, EAT-C trains one planning policy to decompose a hard task into sub-tasks of multiple difficulty levels by only using the RL's time cost data. It directly generates the sub-tasks and thus requires neither hierarchical partition of the task space nor a predefined set of sub-tasks. Compared to HP and HRL, EAT-C (1) enables a mutual training between the path-planner and the RL agent; (2) has an adversarial EG to adjust each sub-task's environment to be more challenging; and (3) improves the training efficiency under the guidance of easy-to-hard curricula automatically generated for each stage.

## 3 PROBLEM FORMULATION

### 3.1 GOAL-CONDITIONED REINFORCEMENT LEARNING

Goal-conditioned RL or multi-goal RL learns a policy that can adapt to different goals. Given the state space $\mathcal{S}$, the action space $\mathcal{A}$, and the goal space $\mathcal{G}$, a goal-conditioned policy is a mapping $\pi(a|s,g) : \mathcal{S} \times \mathcal{G} \mapsto \mathcal{A}$ that outputs an action $a$ (or probabilities $\Pr(a|s,g)$ over actions $a \in \mathcal{A}$) given a state-goal pair $(s,g)$. An RL agent starts from a initial state $s = s_0$ and uses $\pi(a|s,g)$ to take a sequence of actions and interact with the environment, which determines the agent's new state and reward after the action taking in each step. The interaction with the environment is defined by a Markov decision process (MDP) $\{\mathcal{S}, \mathcal{A}, \mathcal{G}, p, r, \gamma\}$, where $p(s'|s,a) \triangleq \Pr(s_{t+1} = s'|s_t = s, a_t = a)$ is the transition probability for the agent from state $s$ to $s'$ after taking action $a$, $r(s, a|g) : \mathcal{S} \times \mathcal{A} \times \mathcal{G} \mapsto \mathbb{R}$ is a reward function, and $\gamma \in [0,1]$ is a discount factor. For example, it is common to define $r(s,a|g) \triangleq \mathbf{1}\{d(s,g) \leq \epsilon\}$, where $\mathbf{1}$ is the indicator function and $d(\cdot,\cdot)$ is a distance metric, so the agent achieves reward of 1 if reaching an $\epsilon$-neighborhood of the goal $g$.

The learning objective of goal-conditioned RL is to maximize its expected return over different tasks $(s_0, g)$, i.e., $\max_\pi \mathbb{E}_{(s_0,g)}[\mathbb{E}_\pi(R_0)]$ where the return at step-$t$ is defined as $R_t = \sum_{i=t}^T \gamma^{i-t} r(s_t, a_t|g)$. This is usually used as the objective for policy gradient methods. Define the action-value function $Q(s,a|g) \triangleq \mathbb{E}(R_t|s_t = s, a_t = a, g)$, the optimal policy $\pi^*$ achieves the maximal $Q(s,a|g)$ for any feasible $(s,a,g)$. Define the value function $V(s|g) \triangleq \mathbb{E}(R_t|s_t = s, g) = \mathbb{E}_{a\sim\pi}[Q(s,a|g)] = \sum_{a\in\mathcal{A}} \pi(a|s,g)Q(s,a|g)$. To reduce the variance of $R_t$, Actor-critic methods additionally learns a model of $V$ or $Q$ as a "critic" to the "actor" $\pi$. Their training objectives aim to minimize the Bellman residual, i.e., the difference between the two sides of Bellman equation:

$$Q(s_t, a_t|g) = r(s_t, a_t|g) + \gamma\mathbb{E}_{s_{t+1}\sim p}[\mathbb{E}_{a\sim\pi}[Q(s_{t+1}, a|g)]]. \tag{1}$$

In experiments, to encourage exploration, we use soft actor-critic (SAC) (Haarnoja et al., 2018b).

### 3.2 GENERATING SUB-TASK CURRICULA FOR GOAL-CONDITIONED RL

Training goal-conditional RL on long-horizon tasks can be highly inefficient since sparse rewards can only be achieved after taking a long sequence of actions, which cannot provide informative feedback to improve the policy. Moreover, the immature policy in earlier stages can easily fail and make the rewards even more sparse. In this paper, we train **a path-planning policy** $\pi_p$ to recursively decompose a long-horizon task into coarse-to-fine sequences of easier sub-tasks as a sub-task tree. Solving any of these sequence can accomplish the original task but the finer ones composed of more sub-tasks provide denser rewards to RL. Hence, the path-planner generates an easy-to-hard curriculum to train the RL agent, which starts from learning easier sub-tasks in finer sequences and gradually focus on more challenging sub-tasks in coarser ones. However, some sub-tasks can still be either too trivial for RL or redundant to other sub-tasks. Moreover, the RL agent is not trained to adapt to small changes in the environment. To address these two issues, we propose **an environment generator (EG)** $\pi_e$ that adversarially modifies the environment of each sub-task to be more challenging and informative for RL. Therefore, the path-planner and the EG are complementary in producing an efficient curriculum of tasks and environments to train the RL agent.

### 3.2.1 CO-OPERATIVE PATH PLANNER: LEARNING EASY-TO-HARD SUB-TASKS

We extend "sub-goal tree (SGT)" (Jurgenson et al., 2020) to generate our curriculum of sub-tasks. Given an long horizon task $(s_0, g)$ with initial state $s_0$ and final goal $g$, we recursively apply a path planning policy $\pi_p(g|g_i, g_j)$ to interpolate sub-goals between $s_0$ and $g$. In particular, given two sub-goals $g_i$ and $g_j$, sampling from $\pi_p(g|g_i, g_j)$ yields a sub-goal interpolated between $g_i$ and $g_j$. Hence, we can generate a sub-goal tree $g_{0:T}$ for $(s_0, g)$ by

$$\Pr_{\pi_p}(g_{0:T}|g_0 = s_0, g_T = g) \triangleq \Pr_{\pi_p}\left(g_{0:\frac{T}{2}}|g_0, g_{\frac{T}{2}}\right) \Pr_{\pi_p}\left(g_{\frac{T}{2}:T}|g_{\frac{T}{2}}, g_T\right) \pi_p\left(g_{\frac{T}{2}}|s_0, g\right), \quad (2)$$

where $T = 2^K$ with $K$ being the depth of the tree. As shown in Fig. 1, layer-$k$ of the sub-goal tree $g_{0:T}$ interpolate a sequence of $2^k - 1$ sub-goals $g^k_{1:2^k-1} \triangleq \left(g^k_1, g^k_2, \cdots, g^k_{2^k-1}\right)$ between $s_0$ and $g$, where $g^k_j = g_{Tj/2^k}$ in $g_{0:T}, \forall j \in \left[2^k - 1\right]$. The goal of path planning is to generate cost-efficient sub-tasks for the RL agent, so we train $\pi_p$ by minimizing the time cost $c(g^k_{0:2^k})$ of the sub-goal trajectory $g^k_{0:2^k}$ on each layer-$k$, i.e.,

$$\min_{\pi_p} J_{\pi_p} \triangleq \mathbb{E}_{(s_0,g)} \mathbb{E}_{g_{1:T-1} \sim \pi_p} \left[\sum_{k=1}^{K} c(g^k_{0:2^k})\right], \quad c(g^k_{0:2^k}) \triangleq \sum_{t=0}^{2^k-1} c\left(g^k_t, g^k_{t+1}\right), \quad (3)$$

where $c(g^k_t, g^k_{t+1})$ represents the time-cost that the agent taken from $g^k_t$ to $g^k_{t+1}$.

### 3.2.2 ADVERSARIAL-ENVIRONMENT GENERATOR (EG) APPLIED TO EACH SUB-TASK

Given the next sub-task $(g^k_t, g^k_{t+1})$ in layer-$k$, EG policy $\pi_e$ adversarially modifies the environment $E^k_{t-1}$ of previous sub-task $(g^k_{t-1}, g^k_t)$ to be more challenging to the RL agent, i.e., sampling subtask-$t$'s environment $E^k_t \sim \pi_e(E|s^k_t, g)$ where $s^k_t \triangleq (E^k_{t-1}, g^k_t, g^k_{t+1})$ denotes the state of EG at subtask-$t$. As an adversary to the RL agent, the reward function for EG is defined as $r_e(s^k_t, E^k_t|g) \triangleq -\mathbf{1}\{r(s_t, a_t|g) = 1\}$ where $(s_t, a_t)$ refer to the state-action of the RL agent at the end of subtask-$t$. Thereby, EG receives the minimum reward $-1$ when the RL agent successfully finishes subtask-$t$ and otherwise the reward is $0$. We can also define an MDP for EG, which mainly differs from the RL agent in that each step corresponds to a sub-task so EG is only allowed to modify the environment at the beginning of each sub-task.

Similar to goal-conditioned RL defined in Sec. 3.1, the learning objective of EG is to maximize its expected return over different tasks $(s_0, g)$, i.e., $\max_{\pi_e} \mathbb{E}_{(s_0,g)}[\mathbb{E}_{\pi_e}(R^e_0)]$ where the return is defined as $R^e_t = \sum_{k=1}^{K} \sum_{i=t}^{2^k} \gamma_e^{i-t} r_e(s^k_t, E^k_t|g)$ with discount factor $\gamma_e \in [0, 1]$. By defining the corresponding value function $V_e$ and action-value function $Q_e$ as in Sec. 3.1, we can apply any RL algorithm to train EG, e.g., we use A2C (Mnih et al., 2016) for the experiments.

## 4 ENVIRONMENT-ADVERSARIAL SUB-TASK CURRICULUM

### 4.1 AUTO CURRICULUM GENERATION AND MUTUAL-BOOSTING

In EAT-C, we need to jointly train three policies, i.e., the path-planner $\pi_p$ and EG policy $\pi_e$ that generate tree-structured curricula of sub-tasks, and the RL policy $\pi$ to accomplish the targeted tasks. At the first glance, training three policies can be more difficult than training one RL policy and requires to collect more data via interactions. Moreover, it is challenging to directly train a path-planner generating dense sub-goals and EG can also suffer from sparse rewards on long-horizon tasks. However, EAT-C allows the three policies help each other's training via a mutual boosting mechanism, where each policy is progressively trained on a curriculum of easy-to-hard sub-tasks using dense feedback from other policies on the sub-tasks. By iterating this mutual-boosting on sub-task curricula, EAT-C significantly improves the training efficiency of each policy and results in an RL agent with better generalization to unseen tasks and perturbed environments.

In each episode, we train the path-planner during its "top-down" construction of a sub-task tree: it starts from learning to interpolate a few sub-goals between the given task $(s_0, g)$ and gradually moves to more challenging cases of generating dense sub-goals in bottom layers of the tree. Since it aims to generate the most cost-efficient path of sub-goals for the RL agent, we use the time cost of the RL agent on those sub-tasks in the previous episode as training data, which provide dense rewards to accelerate the training of the path-planner.

Given a constructed sub-task tree, we then train the RL policy and EG policy by a "bottom-up" traversal of the sub-task tree, which naturally forms an easy-to-hard curriculum for each policy. Specifically, both policies firstly learn from dense rewards by finishing easier sub-tasks in bottom layers, where the RL agent only needs to reach a nearby sub-goal and the EG is allowed to frequently modify the environments between $(s_0, g)$. Due to the adversarial training between them, they are not only learning from the environments but also from each other. As moving to the top layers, the RL agent receives less guidance from fewer sub-tasks, while the EG can only change the environment once in each long-horizon sub-task. Thereby, they need to improve their policies learned on easier tasks for more challenging tasks. As a result, the RL agent learns to adapt to different tasks and perturbed environments, while the EG learns to generate more powerful adversarial perturbations to train the RL agent in the next episode.

In EAT-C, for each long-horizon task, we iterate the above mutual-boosting training on new sub-task curricula re-generated by the updated path-planner and EG for multiple episodes. This is an imitation of human learning that repeatedly practicing the same complicated task in different ways (e.g., different sub-tasking and perturbed environments). The path-planning and adversarial modification of environments are complementary in constructing a curriculum for more efficient RL: the former decomposes a hard task into easier sub-tasks while the latter modifies them to be sufficiently challenging and diverse so the RL agent can learn different skills with better generalization to unseen tasks or perturbed environments.

## 4.2 EAT-C ALGORITHM

**Top-Down Planning of Sub-task Curriculum** We provide the detailed procedure of the top-down construction of the sub-task curriculum and the update of path-planner $\pi_p$ in Algorithm 2 (Appendix. C). For each layer-$k$ from $k = 0$ to $k = K$, EAT-C firstly updates the planning policy $\pi_p$ in line 4 by an RL algorithm using the time cost data collected on sub-tasks up to layer-$k$ from the previous episode's bottom-up training (i.e., Algorithm 3, Appendix C), and then recursively generates the sub-tasks on

---

**Algorithm 1** EAT-C

1: **Input:** $p_0, T, \tau_{\max}, \epsilon, n$
2: **Output:** RL agent's policy $\pi$, planning policy $\pi_p$, EG policy $\pi_e$
3: **Initialize:** $\pi, \pi_p, \mathcal{D}_p$
4: **while** not converge **do**
5:     Sample a task $(s_0, g)$ with $s_0 \sim p_0$ and $g \in \mathcal{G}$;
6:     **for** episode $= 1, 2, \ldots, n$ **do**
7:         **Algorithm 2**: top-down construction of a sub-task tree $g_{0:T}$, train planning policy $\pi_p$ based on $\mathcal{D}_p$;
8:         **Algorithm 3**: bottom-up traversal of the sub-task tree $g_{0:T}$, train RL policy $\pi$ and EG policy $\pi_e$, collect $\mathcal{D}_p$;
9:     **end for**
10: **end while**

---

layer-$k$ for the current episode (line 5-8). Hence, the path-planner firstly learns to plan coarse trajectories of fewer sub-goals in top layers and gradually increases the sub-goals to form finer paths that can provide more guidance to RL in bottom layers. Since we always use the latest time cost data from $\mathcal{D}_p$ for training, the planning policy $\pi_p$ keeps being optimized to generate cost-efficient sub-task trajectories for the latest RL policy $\pi$.

**Bottom-Up Curriculum for RL and EG** Given a sub-task tree generated by Algorithm 2, EAT-C trains RL policy $\pi$ and EG policy $\pi_e$ by following a bottom-up traversal of the tree. As shown in Algorithm 3, it starts from learning easier sub-tasks on the bottom layer-$K$ and gradually moves to top layer-0 (line 4-13), which recovers the original long-horizon task. In each layer, we reset the initial state of the RL agent (line 5) and apply $\pi_e$ and $\pi$ to a sequence of $2^k$ sub-tasks $g_{0:2^k-1}^k$ towards the final goal $g$ (line 6-11), and then we update the two policies using the experiences collected on these sub-tasks (line 12). On each sub-task, EG adversarially perturbs the environment (line 7) and the RL agent is then applied to accomplish this modified sub-task (line 8). If the RL agent fails and ends at a state $s$, EAT-C recursively invoke the planning policy $\pi_p$ to add more sub-goals between $s$ and the sub-task's goal to provide more detailed guidance to the RL agent until it reaches the goal. This is described in line 9 and line 14-21.

**EAT-C algorithm** The complete procedure of EAT-C is introduced in Algorithm 1. Given a long-horizon task $(s_0, g)$ (line 5), EAT-C iterates between the top-down and bottom-up procedures in Algorithm 2-3 for $n$ episodes (line 6-9) before moving to a new long-horizon task. Therefore, the planning policy $\pi_p$ and EG policy $\pi_e$ are optimized to produce better curricula of sub-tasks to train the RL agent to complete task $(s_0, g)$ while the RL agent learns skills for solving different sub-tasks and generalizing to non-trivial changes of the environment.

## 5 EXPERIMENTS

In this section, we evaluate EAT-C and compare it with a broad range of RL methods on two benchmarks, i.e., navigation and manipulation of a 6-point 2D robot to push an object to a goal state, and three compositional tasks in a discrete space. In these experiments, we mainly focus on their efficiency on long-horizon tasks and generalization to environments with small changes. Moreover, we present an ablation study to evaluate the contribution of each part in EAT-C. In addition, we provide case studies to analyze the planned sub-task tree and the modified environment for each sub-task, which explains why EAT-C can improve RL in several aspects. Our experiments are conducted on two representative RL benchmarks and a diverse set of baselines:

**2D pusher** (Yamada et al., 2020). As shown in Fig. 4, this is a robot navigation and manipulation task in a continues space: a 2D robot with a 4-joint arm needs to firstly navigate to an object and then push it to a goal location within an environment of multiple obstacles. We train EAT-C on 50 environments with the positions and sizes of obstacles randomly sampled from uniform distributions (more details in Appendix A). In each environment, we randomly sample 80 training tasks with different $(s_0, s_{obj}, g)$, where $s_0$ is the initial state, $s_{obj}$ is the state of the object, and $g$ is the goal state. Most evaluated policies in our experiments need $\geq 250$ steps to finish each task so they are long-horizon tasks. The rewards are sparse since the agent only receives a reward when near the object or when pushing the object and reaching the goal. In EAT-C, the path-planner generates sub-tasks between $(s_0, s_{obj})$ and $(s_{obj}, g)$. For every sub-task, EG can perturb the sizes and locations of at most three obstacles within pre-defined ranges. For the test, we randomly sample 30 new environments each having one randomly sampled task.

**Discrete space tasks** (Maxime Chevalier-Boisvert & Pal, 2018). We train and test RL policies on three types of compositional tasks depicted in fig 3, i.e., hunting, scavenging, and salad-making, as illustrated in Fig. 3.(a)-(c). The agent needs to take two or three key steps to finish each task and only get reward when finishing each key step. There are 250 environments used for training and 100 environments for test, each associated with one task. In EAT-C, the path-planner is applied to every two consecutive key steps. EG perturbs the environment by moving objects including tree and stones.

**Baselines:** We compare EAT-C with a broad class of RL methods having related ideas to EAT-C: (1) ALP-GMM (Portelas et al., 2019) that generates a curriculum of diverse tasks with large progress for goal-conditioned RL; (2) POET (Wang et al., 2019) with two auxiliary agents for generating a curriculum of adversarial yet solvable environments to accelerate RL; (3) Ecological RL (Co-Reyes et al., 2020) that dynamically modifies the environment to improve non-episodic (and thus long-horizon) RL without reset of the initial state; (4) A hierarchical RL method (Zhang et al., 2021); (5) Zhang et al. trains the RL agent by modelling a goal proposal curriculum that samples goals at the frontier of the set of goals that an agent is able to reach. All these baselines and EAT-C need to invoke an RL algorithm as their subroutine. For fair comparisons, we use Soft Actor Critic (SAC) (Haarnoja et al., 2018a) in all evaluated methods for its stable and promising performance. All methods are not allowed to modify the environments during test since test environments are assumed to be the realistic ones in which we deploy the RL agents. We run different methods for the same number of interaction steps with the environment, i.e., $20 \times 10^6$ for 2D pusher and $1.25 \times 10^6$ for discrete space tasks. More details of experimental settings are given in Appendix A.

### 5.1 MAIN RESULTS

We report the performance of EAT-C and all baselines evaluated on the test tasks in Fig. 2(a) for 2D pusher and in Fig. 3(d)-(f) for the discrete space tasks. In all experiments, EAT-C outperforms all other baselines by a large margin on both the learning efficiency and the final generalization performance to test tasks in new environments. In most experiments, baselines adopting a curriculum of environments, i.e., ALP-GMM, POET, and Ecological RL, performs worse than EAT-C but better than the other baselines without changing environments for training. This indicates that building a curriculum of training environments is essential to improving RL's generalization and robustness to small changes in the deployed environments. Moreover, among the three compositional tasks in Fig. 3, hunting and scavenging contain moving object, i.e., the deer and the predator, which require the RL agent to adapt to the changes of their locations. On these two tasks, EAT-C exhibits more advantages over other baselines than on the salad-making task, which does not contain any moving object. Therefore, EAT-C enables RL to efficiently learn to adapt to changes in the environment.

By comparing the methods with environment changed, we notice that controlling the difficulty of the modified environments is critical to earlier-stage learning efficiency. In particular, both EAT-C and POET trains another agent (e.g., the path-planner in EAT-C and the antagonist agent in POET) to control the hardness of the training tasks matching the capability of the RL agent, so their earlier-stage improvement than others. In contrast, the environments selected in ALP-GMM might be too challenging and the ones modified by POET might be too easy, leading to poorer efficiency in earlier stages.

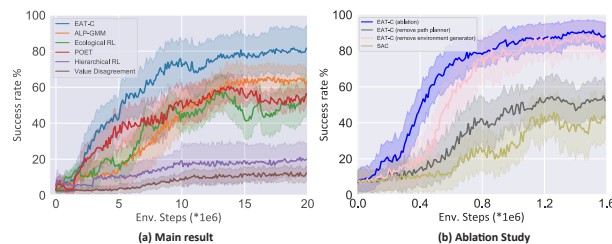

Figure 2: **(a)** report the success rate (mean±std averaged over 6 random seeds) of EAT-C and baselines on test tasks in 2D Pusher environments. **(b)** Ablation study of EAT-C on 2D Pusher tasks.

Although these methods are designed to train RL policies that can adapt to different environments or tasks, only EAT-C trains a path-planner to decompose a long-horizon task into a curriculum of easy-to-hard sub-task sequences for training. The guidance of path-planner and its curriculum plays a critical role in outperforming these baselines.

## 5.2 ABLATION STUDY

EAT-C jointly trains a path-planner and an environment generator (EG) to produce a curriculum of sub-tasks to improve RL. Hence, we conduct a thorough ablation study of how the performance changes if removing each component from EAT-C. In particular, we compare the original EAT-C with three variants, i.e., EAT-C with path-planner removed, EAT-C with EG removed, and SAC (with both removed), on the 2D Pusher tasks. Since the last two variants are not trained on perturbed environments, for fair comparisons, we use the same environment for both training and test and only create new test tasks by sampling $(s_0, s_{obj}, g)$. The test performance is reported in Fig. 2(b).

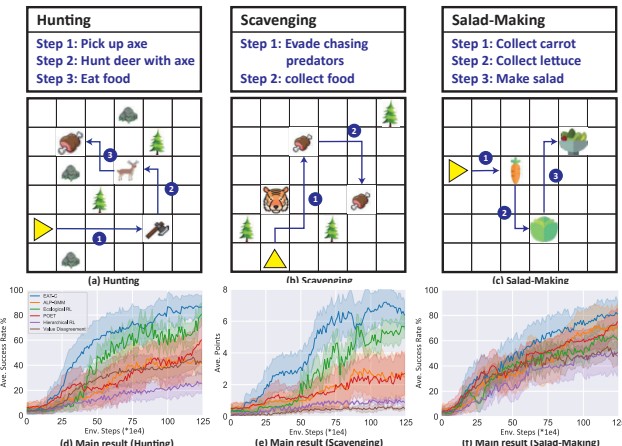

Figure 3: **(a)-(c)** illustrate the 2-3 key steps for completing each task. In Scavenging, the agent will have 2 points when it collects food each time. **(d)-(f)** report different methods' performance (mean±std over 10 random seeds) on multiple test tasks.

When the path-planner is removed from EAT-C, we no longer have any easy-to-hard curriculum of sub-tasks to train the RL agent or EG and they can only learn inefficiently from the original long-horizon tasks. The adversarial environments generated by EG make tasks for RL even harder and unsolvable. Hence, we can observe that it only completed 50% of the test tasks within $1.6 \times 10^6$ environment steps, compared to 90% of the original EAT-C. This indicates that the plan-planner and its generated sub-task tree are critical in creating an effective curriculum for RL.

When EG is removed from EAT-C, we still have the curriculum of sub-tasks to train the RL agent but some sub-tasks might be too trivial or redundant to provide informative feedback for improving the RL agent. This results in poorer efficiency in earlier-stages compared to the original EAT-C with an EG: it reaches 80% success rate after $0.84 \times 10^6$ interaction steps instead of $0.73 \times 10^6$ steps required by the original EAT-C. Although ETA-C without EG can eventually achieve a comparable success rate at the end of training. During later stages, the success rate fluctuates unstably over time, while EAT-C with an EG performs more robustly due to the adversarial environments used for training. Moreover, EAT-C significantly improves SAC by a large margin via running SAC on an automatically generated curriculum of sub-tasks, which implies the importance of curriculum on improving RL's efficiency.

## 5.3 HOW DOES EAT-C WORK? AN EMPIRICAL STUDY

To better understand how the path planner and EG help RL in EAT-C, in Fig. 4, we visualize the sub-task tree (with layer $k \in \{0, 1, 2\}$) generated by the path-planner and the EG's modifications to the environment in each sub-task at Episode 3 and Episode 6 (episode was defined in Alg. 1) for a 2D

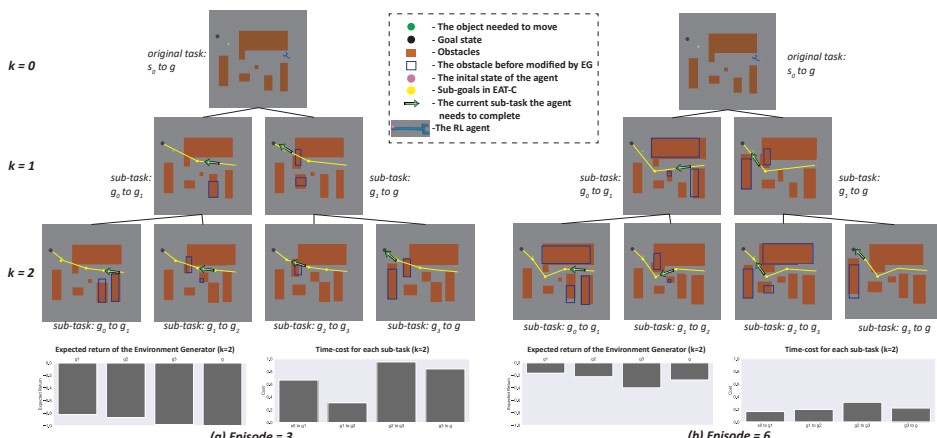

Figure 4: Visualization of EAT-C. A 2D robot with a 4-joint arm starts from the initial state (pink), navigates to the object (green) location, and then pushes the object to the goal state (black). The histograms in (a) and (b) represents the expected return of EG taking action $b_t$, and the costs of sub-tasks predicted by the path planner in layer $k = 2$, respectively.

Pusher task $(s_0, s_{obj}, g)$. In the histograms, we also report the expected return of EG and the time cost of the RL agent on each sub-task from the bottom layer $k = 2$ for the two episodes.

**The curriculum of sub-tasks generated by the path-planner**. Each plot on the tree describes a sub-task, where the arrow highlights the sub-task and the yellow trajectory denotes the sequence of all sub-tasks of the layer. Comparing the sub-tasks in different layers, bottom layers (e.g., $k = 2$) provides more guidance and dense rewards to the RL agent while the sub-tasks in upper layers (e.g., $k = 1$) are much harder. Comparing the same-layer sub-tasks generated in different episodes, the sub-tasks in Episode 3 do not take all obstacles into account, e.g., some sub-task sequences trespass obstacles and some sub-tasks are too close to obstacles, because the planning policy is not fully optimized yet to produce a cost-efficient path for the RL agent. Hence, the time costs for the RL agent to accomplish these sub-tasks can be much higher than later episodes. Moreover, the time costs of some sub-tasks can be much higher than others and thus cannot provide dense rewards to assist RL. On the contrary, in Episode 6, the path-planner has learned to generate cost-efficient sub-tasks with similar hardness so the trajectories and sub-goals are distant from the obstacles and can provide dense rewards facilitating RL. Comparing the histograms of time costs for the two episodes, the RL agent is significantly improved by learning to complete the sub-tasks in the easy-to-hard (bottom-up) curriculum.

**Adversarial modifications to obstacles in the environments:** In each sub-task plot of Fig. 4, EG adversarially modifies some obstacles by changing their previous sizes and positions (depicted by the blue boxes) to make the sub-tasks sufficiently challenging and diverse. Similar to the path-planner, EG is improved over time: for example, in the sub-task "$g_0$ to $g_1$" on layer-2, the RL agent needs to pass a corridor formed by three obstacles, while EG makes the corridor longer and narrower and thus more challenging for the agent to pass in Episode 6, its modification in Episode 3 is not ideal and even moves one obstacle away from the agent. The improvement of EG is also reflected by its increasing expected return shown in the two histograms. By modifying the environments to be more difficult in sub-tasks, EG encourages the RL agent to learn diverse skills in different sub-tasks. Hence, the sub-tasks are easy for the agent to collect dense rewards but they are non-trivial and informative because of EG's modifications.

## 6 CONCLUSION

We propose a mutual learning and auto-curriculum framework "EAT-C" to improve the efficiency of RL on long-horizon tasks as well as its generalization and robustness to new environments. EAT-C trains a planner to decompose a hard task into coarse-to-fine sequences of sub-tasks providing an easy-to-hard curriculum to train an RL agent, while an adversarial environment generator modifies these sub-tasks to be diverse and more informative to learn. The three policies are trained with data collected by each other. On two types of tasks, EAT-C outperforms a diverse set of baselines, e.g., curriculum-based RL, hierarchical RL, and planning-based methods. It has the potential to be applied to more complicated tasks with dynamic environments or visual inputs such as games, which will be covered in our future works.

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

## A    DETAILS OF ENVIRONMENT IMPLEMENTATIONS

### A.1    2D PUSHER

We simulate the environment of 2D pusher in Mujoco physics engine (Todorov et al., 2012). The positions of the robot, object and goal are defined as $p_{rob}, p_{obj}$, and $p_{goal}$, respectively, and $T$ is the maximum number of episodes (i.e., the horizon). An 2D-pusher agent with four joints can take actions of six dimensions, two for navigation and the rest four for arm control. The map size is $1 \times 1$ so both the x and y coordinates lie in $(-0.5, 0.5)$. The x and y coordinates of goals and objects are randomly sampled from uniform distributions of $\mathcal{U}(-0.35, -0.2)$ and $\mathcal{U}(-0.15, 0.1)$, respectively. The initial state of the agent is randomly sampled from uniform distribution of $\mathcal{U}(-0.05, 0.3)$. For obstacles, their initial coordinates and sizes are randomly drawn from an uniform distribution, as explained in the first row of Table. 1. We train 2D pusher using sparse reward: when the robot reaches a vicinity of the object or the sub-goal state ($||p_{rob} - p_{obj}||_2 \leqslant 0.05$) the agent will receive a reward$= 150/2^k$, where $k$ is the layer of the sub-task tree where the sub-goal is located. Once the agent pushes the object to the goal state with $||p_{goal} - p_{obj}||_2 \leqslant 0.05$, the agent will receive a one-time reward$= 150$; otherwise there is no reward. By taking an action, EG can change the size and the location of $0 \sim 3$ obstacles near the agent. Assume that there are $n$ obstacles in the environment, and we represent each obstacle-$i$ by its location $(x_i, y_i)$ (2D coordinates) and size $(w_i, h_i)$ (width and height) as $\theta_i = (x_i, y_i, w_i, h_i)$. The action $b_t$ of EG is defined as

$$b_t \triangleq \Delta\theta_i = (\Delta x_i, \Delta y_i, \Delta w_i, \Delta h_i), \ \forall i \in [n]. \tag{4}$$

In order to provide feasible and smoothly changing environments to RL along the sub-task trajectory in each layer, and to prevent the environment generator from being too powerful and overly adversarial, it is important to restrict EG from changing the environment too much at one time for each sub-task. Hence, in the experiments, we constrain every dimension in an action of EG not to exceed some threshold, e.g., for $x_i$, we apply

$$\Delta x_i \leftarrow \min\{\max\{\Delta x_i, \beta_x \cdot x_{\min}\}, \beta_x \cdot x_{\max}\}, \tag{5}$$

where $\beta_x \in [0, 1]$ and $(x_{\min}, x_{\max})$ is the valid range of x-coordinate. The above environment parameterization can be easily extended to other environments so EAT-C is a general and principal scheme that can be adapted to different environments.

### A.2    DISCRETE SPACE TASKS

The environment for the three tasks is an $N \times N$ grid. It is partially observed by the RL agent: the agent at each state obtains a local egocentric view of a $5 \times 5$ grid around it, where the object in each cell of the grid is represented by a $C$-dimensional one-hot vector (there are $C$ possible types of objects). The agent can pick up and carry one object at a time. It can also combine two objects to construct a new one by putting a carried object onto an existing object, e.g., it can combine wood with metal to make an axe. The RL agent can take action such as moving in the cardinal directions, picking up an object, and dropping an object. In discrete space tasks, the environment generator (EG) can modify the environment by taking an action to move an object/obstacle. In order to provide feasible and smoothly changing environments to RL along the sub-task trajectory, and to prevent EG from being too powerful and overly adversarial, each action of EG can only move every object/obstacle to an adjacent cell around it.

### A.3    ENVIRONMENT ENCODING OF BASELINES

The baselines in our experiments include two curriculum RL methods, i.e., ALP-GMM and POET, which both can control some environment-dependent parameters for mutation and generation of the training environments. For their fair comparisons to EAT-C, we set their environment-dependent parameters exactly the same as the ones controlled by EG in EAT-C. For 2D-pusher, as detailed in Table 1, the baselines control the same four environment-dependent parameters as EAT-C. Each parameter is initialized by sampling from a uniform distribution and each mutation step can modify it by a small value if it is within the valid range. For the discrete space tasks, ALP-GMM/POET can generate environments by sampling/mutating the location of each obstacle/object, which is the same parameter controlled by EAT-C. The valid range for location for each obstacle/object is $(0, 1)$ and the mutation step size is $0.1$.

Table 1: Environment-dependent parameters in baselines on 2D-pusher. Each baseline generates an environment of obstacles by uniformly sampling the four parameters defining each obstacle from the corresponding ranges. It starts from the initial ranges below and can take a mutation step one time to change the lower and upper bounds of each parameter's range, if the two bounds do not exceed their minimal and maximal values listed below. The two numbers in each tuple $(\cdot, \cdot)$ below corresponds to the lower and upper bound of the range.

| Parameter Type | Obstacle x-coordinate $x_i$ | Obstacle y-coordinate $y_i$ | Obstacle width $w_i$ | Obstacle height $h_i$ |
|---|---|---|---|---|
| Initial Range | $(-0.3, 0.3)$ | $(-0.3, 0.3)$ | $(0, 0.1)$ | $(0, 0.1)$ |
| Mutation Step | $(0.02, 0.02)$ | $(0.02, 0.02)$ | $(0.05, 0.05)$ | $(0.05, 0.05)$ |
| Minimal Value | $(-0.5, -0.5)$ | $(-0.5, -0.5)$ | $(0, 0)$ | $(0, 0)$ |
| Maximal Value | $(0.5, 0.5)$ | $(0.5, 0.5)$ | $(0.3, 0.3)$ | $(0.3, 0.3)$ |

## B  MODEL ARCHITECTURE AND HYPERPARAMETERS OF SAC (RL ALGORITHM USED IN ALL EXPERIMENTS)

We use the same neural network architecture (i.e., an MLP) for the RL agent and the same RL algorithm (SAC) in all the experiments of all the methods.

Besides the reward of completing a task/sub-task, it is common in MuJoCo and many other simulators to also issue a small instantaneous reward after taking any action in order to encourage exploration. Moreover, different methods usually need to re-scale this exploration reward because they may need different levels of exploration. In our experiments, we tune the re-scaling factor for every method to get its best performance. Specifically, we chose $0.3$ for EAT-C/ALP-GMM/ POET and $8.0$ for hierarchical RL/value disagreement/Ecological RL. An explanation of applying a smaller factor for the former three methods is that they already have some strong exploration strategies and a larger factor might downweigh the task reward and thus results in performance degeneration.

Moreover, we use the same coefficient $\alpha$ of the entropy term in SAC's objective for all methods (they all use SAC as the RL algorithm). The coefficient $\alpha$ controls the degree of exploration and is automatically tuned. A complete list of hyperparameters for SAC in 2D-pusher tasks is given in Table 2. They are exactly the same hyperparameters defined in in SAC paper (Haarnoja et al., 2018a) and in Table 1 of their Appendix D except that we choose different values for them in 2D-pusher.

In the discrete space tasks, the environment is a $10 \times 10$ grid and the $5 \times 5$ partial observation (as mentioned in A.2) of the RL agent can be represented as a $5 \times 5 \times C$ one-hot tensor. We flatten this tensor to a vector and process it by an MLP with three hidden-layers whose output dimensions are $(64, 64, 32)$, respectively. We apply another MLP with three layers of output dimensions $(16, 16, 16)$ to process the inventory observation. The two MLPs' outputs are then concatenated and processed by an MLP with two hidden layers of output dimensions $(16, \text{action\_dimension})$ that outputs a probability distribution over all possible actions. We use ReLU as our nonlinear activation functions in all MLP models except their last layer, which uses a softmax function to compute the probability of taking each action. In EAT-C, the RL agent and EG share the same observations as well as the first two MLP models but they use different MLP models to output the actions. A complete list of hyperparameters of SAC in the discrete space tasks is given in Table. 3.

## C  PSEUDO-CODE OF EAT-C

In the training phase, we first randomly initialize $\pi_p$, and apply it to predict a sub-task tree using Euclidean distance as an initialization of the cost in line 4 of Algorithm 2 when no time cost data have ever been collected at the very beginning. Given the sub-task tree, we can train the RL agent by a bottom-up curriculum of the sub-tasks on the tree. In particular, we start from the bottom layer and train the RL agent to consecutively complete a sequence of sub-tasks from the starting state to the goal state. As training proceed, the RL agent collects data of the time cost for completing feasible sub-tasks $(g, g')$. When the RL agent cannot complete the pre-assigned sub-task $(g_i^k, g_{i+1}^k)$ within a time limit, we re-apply $\pi_p$ to interpolate more sub-goals between $(g_i^k, g_{i+1}^k)$ by line 25 in Algorithm 3.

Table 2: SAC hyperparameters in EAT-C (2D-pusher)

| Parameter | Value |
|---|---|
| Optimizer | Adam |
| Learning rate | $3.0 \times 10^{-4}$ |
| Discount factor $(\gamma)$ | 0.99 |
| Replay buffer size | $1.0 \times 10^{6}$ |
| Number of hidden layers for all networks | 2 |
| Number of hidden units for all networks | 400 |
| Minibatch size | 256 |
| Nonlinearity | ReLU |
| Target smoothing coefficient $(\tau)$ | $5.0 \times 10^{-3}$ |
| Target update interval | 1 |
| Network update per environment step | 1 |
| Entropy target | $-\dim(\mathcal{A})$ |

Table 3: SAC hyperparameters in EAT-C (discrete space tasks)

| Parameter | Value |
|---|---|
| Optimizer | Adam |
| Learning rate | $5.0 \times 10^{-4}$ |
| Discount factor $(\gamma)$ | 0.99 |
| Replay buffer size | $1.0 \times 10^{6}$ |
| Number of hidden layers for all networks | 3 |
| Number of hidden units for all networks | 256 |
| Minibatch size | 256 |
| Nonlinearity | ReLU |
| Target smoothing coefficient $(\tau)$ | $5.0 \times 10^{-3}$ |
| Target update interval | 1 |
| Network update per environment step | 1 |
| Entropy target | $-\dim(\mathcal{A})$ |

---

**Algorithm 2** Top-Down Planning of Sub-task Curriculum

---

1: **Input:** $(s_0, g)$, $T$, planning policy $\pi_p$ and its training set $\mathcal{D}_p$.
2: **Output:** tree structured sub-goals $g_{0:T}$, $\pi_p$
3: **for** $k = 1, 2, \ldots, K$ **do**
4:     Apply an RL algorithm to minimize $J_{\pi_p}$ in Eq. (3) computed on time cost data up to layer-$k$ in $\mathcal{D}_p$;
5:     **for** $t = 1, \ldots, 2^{k-1}$ **do**
6:         Generate the sub-goal $g_t^k \sim \pi_p(g_t^k | g_{t-1}^{k-1}, g_t^{k-1})$;
7:         Add $g_t^k, g_t^{k-1}$ into the trajectory $g_{0:T}^k$ on layer-$k$;
8:     **end for**
9: **end for**

---

More technical details are given in Algorithm 3. In the test phase, we apply $\pi_p$ to produce a sub-task tree and only use the sub-task sequence in the bottom layer to guide the RL agent.

# D ADDITIONAL EXPERIMENT RESULTS

## D.1 LARGER VERSIONS OF PLOTS FOR MAIN RESULTS

Considering that the figures of experiment results in the main paper might be too small to read, we temporarily list the main results with a more clear vision (Fig. 5, and Fig. 6).

---

**Algorithm 3** More detailed Bottom-Up traversal in EAT-C

---

1: **Input:** RL policy $\pi$, EG policy $\pi_e$, sub-goal tree $g_{0:T}$, $\tau_{\max}$, $\epsilon$
2: **Output:** $\pi$, $\pi_e$, $\mathcal{D}_p$
3: **Initialize:** $\pi_p$'s training set $\mathcal{D}_p \leftarrow \emptyset$, RL's replay buffer $\mathcal{D} \leftarrow \emptyset$, EG's replay buffer $\mathcal{D}_e \leftarrow \emptyset$
4: **for** $k = K, \ldots, 1, 0$ **do**
5:    Reset RL agent's initial state to $g_0$;
6:    **for** $t = 1, 2, 3, \ldots, 2^k$ **do**
7:       EG adversarially modifies the environment $\mathbf{E}_{t-1}^k$ to $\mathbf{E}_t^k$;
8:       **while** $\tau \leq \tau_{\max}$ or $s_\tau \notin B(g_t^k, \epsilon)$ **do**
9:          RL agent takes action $a_\tau \sim \pi(a_\tau|s_\tau, g_t^k)$;
10:          RL agent moves to $s_{\tau+1} \sim p(s_{\tau+1}|s_\tau, a_\tau)$ and receives reward $r(s_\tau, a_\tau|g_t^k)$;
11:          $\mathcal{D} \leftarrow \mathcal{D} \cup (s_\tau, a_\tau, r(s_\tau, a_\tau|g_t^k), s_{\tau+1})$;
12:       **end while**
13:       REACH$(s, g_{t+1}^k)$;
14:    **end for**
15:    **for** every gradient step **do**
16:       Apply gradient steps in SAC: update $Q$, $V$, $\pi$ using samples drawn from $\mathcal{D}$;
17:       Apply gradient steps in A2C: update $Q_{\pi_e}$, $\pi_e$ using samples drawn from $\mathcal{D}_e$;
18:    **end for**
19: **end for**
20: **Procedure** REACH$(s, g)$:
21: **if** $d(s, g) \leq \epsilon$ **then**
22:    $\mathcal{D}_e \leftarrow \mathcal{D}_e \cup (s_\tau, b_t, r_e(s_\tau, E_t^k|g), g_t^k)$;
23:    $\mathcal{D}_p \leftarrow \mathcal{D}_p \cup (s_0, s_\tau, \tau)$, $s_0 \leftarrow s_\tau$;
24: **else**
25:    Re-apply $\pi_p^i(g_{t-1}^k, g_t^k)$ to predict temporary sub-goals $g'_{1:n}$ for $(g_{t-1}^k, g_t^k)$;
26:    Add $g'_{1:n}$ into the planned sub-goals trajectory $g_{0:2^k-1}^k$;
27:    Re-apply agent start from $s_0$ with the new sub-goal trajectory to reach $g_{2^k-1}^k$ and collect training data (Follow line.8 to line.14);
28:    REACH$(s, g)$;
29: **end if**

---

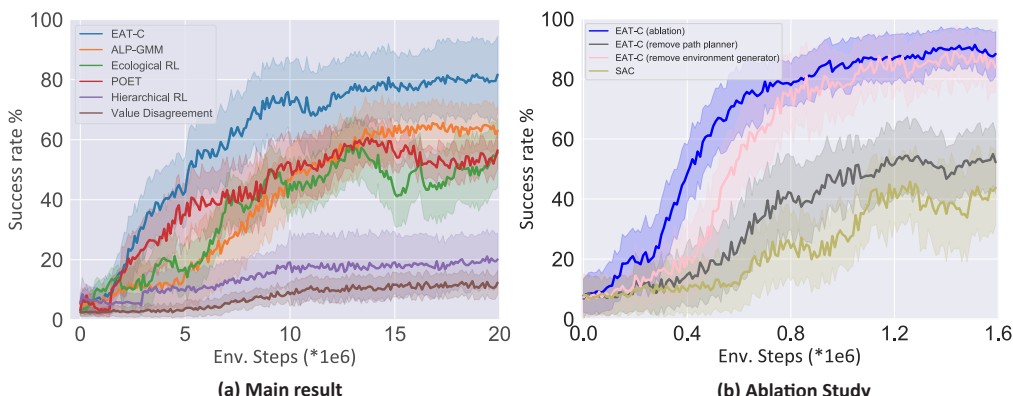

**(a) Main result**      **(b) Ablation Study**

Figure 5: **(a)** report the success rate (mean±std averaged over 6 random seeds) of EAT-C and baselines on test tasks in 2D Pusher environments. **(b)** Ablation study of EAT-C on 2D Pusher tasks.

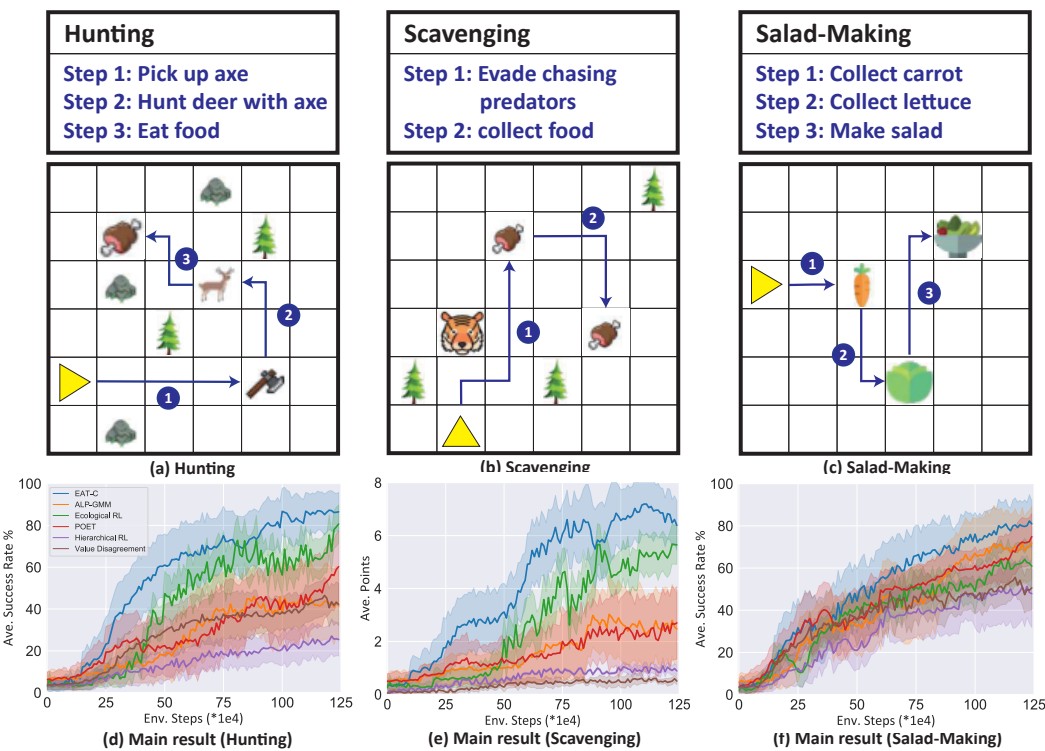

Figure 6: **(a)-(c)** illustrate the 2-3 key steps for completing each task. In Scavenging, the agent will have 2 points when it collects food each time. **(d)-(f)** report different methods' performance (mean±std over 10 random seeds) on multiple test tasks.

## D.2 ADDITIONAL ABLATION STUDY EVALUATING GENERALIZATION/ROBUSTNESS ON NEW RANDOM ENVIRONMENTS

Following the suggestions of Reviewer Zhvq and o38w, we add an ablation experiment and report the results in Table 4. To evaluate the generalization and robustness, which are the advantages of EAT-C due to the adversarial environment generator (EG), we evaluate different methods on **multiple new random environments during the test phase**. This is different from the ablation study in Fig. 2(b), which evaluates all methods on **the fixed training environment**. The new results show a large gap (80.24 vs. 42.23) between EAT-C and EAT-C (remove EG). This demonstrates that EG is important to improving the generalization and robustness of the RL policy. In the training environment (non-random) used in our original ablation study of Fig. 2(b), Hierarchical RL (HRL) does improve the performance of the default RL algorithm (SAC) on long-horizon tasks, i.e., 39.62 (SAC) vs. 68.27 (HRL). However, in random and unseen environments, HRL generalizes much poorer than EAT-C.

Table 4: Ablation Study Results

| Test Setting | Multiple New Random Environments | Training Environment |
|---|---|---|
| EAT-C | 80.24 ± 12.25 | 92.04±6.49 |
| EAT-C (remove EG) | 42.23±10.34 | 85.47±9.12 |
| EAT-C (remove Planner) | 27.58 ± 14.67 | 46.02±10.3 |
| SAC | 20.83 ± 7.24 | 39.62 ± 12.25 |
| Hierarchical RL | 22.04 ± 10.44 | 68.27 ± 6.99 |

Reviewer o38w and Zhvq raised the following concerns: **(1)** whether EG could make the environment more reward sparse? **(2)** whether planner could always generate infeasible sub-goals? To answer these questions, in Fig 7, we report the average time-cost that the agent needs to complete each sub-task in layer-3 of the sub-task tree.

- In earlier stages when $\pi_p$ and the RL agent are not well trained, $\pi_p$ may generate hard sub-tasks. However, after a little training on the sub-task curriculum, $\pi_p$ is trained to generate a minimum-cost path for the RL agent and the capability of the RL agent to finish the sub-tasks is also improved.

- We apply EG to simple sub-tasks that are optimized to be simple in EAT-C (via optimizing the planner) for the RL agent. The goal of EG is to avoid learning similar and easy sub-tasks repeatedly, which cannot provide informative feedback to RL even if the reward is dense. On the other hand, we set several restrictions to avoid over-adversarial environments, as mentioned in Appendix. A

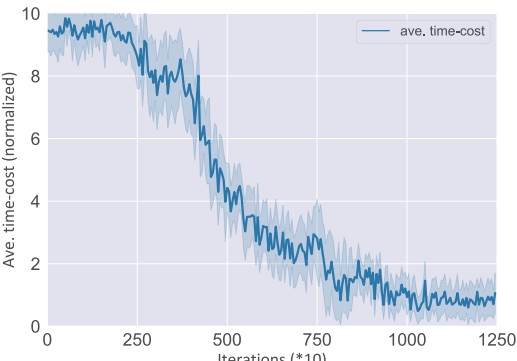

Figure 7: Average time-cost of the RL agent to complete a sub-task from layer-3 of the sub-task tree. As the training proceed, time-cost that the agent needs to complete each sub-task decreases significantly, indicating that $\pi_p$ does not propose infeasible goals, and EG does not make the reward more sparse.

## D.3    EXPERIMENTS ON 7DoF ROBOTIC ARM

Following the suggestions of Reviewer 7T33 and Zhvq, we add an experiment of controlling a 7DoF (degrees of freedom) robotic arm (i.e., the one used in (Jurgenson et al., 2020)) to evaluate how EAT-C performs in a complicated control task. We use MuJoCo as the simulator. In this experiment, the robotic arm learns to avoid obstacles and reach a goal state (as shown in Fig. 8). We compare EAT-C with curriculum RL methods and SGT (Jurgenson et al., 2020).

Both the training and test tasks have 5 obstacles with different and randomly sampled location and size parameters. The start-goal pair of each task are also randomly sampled. Both EAT-C and curriculum RL methods are able to modify exactly the same parameters defining the location and size of each obstacle.

We report the success rate of reaching the goal state without collision and the collision rate as the two metrics to evaluate EAT-C and all the baselines. After training, we evaluate them on 100 new random tasks different from the training tasks. The results are reported in table 5 and EAT-C achieves the best performance on both the collision rate and success rate among all evaluated methods. This demonstrates the robustness and promising performance of EAT-C in more complicated and realistic tasks.

| Methods | Average Collision Rate | Success Rate |
|---------|------------------------|--------------|
| EAT-C   | $\mathbf{0.22} \pm 0.05$ | $\mathbf{0.873} \pm 0.027$ |
| ALP-GMM | $0.34 \pm 0.07$ | $0.524 \pm 0.092$ |
| POET    | $0.42 \pm 0.07$ | $0.544 \pm 0.084$ |
| SGT-PG  | $0.25 \pm \mathbf{0.02}$ | $0.772 \pm \mathbf{0.014}$ |

Table 5: Main results for the experiments on 7DoF robotic arm.

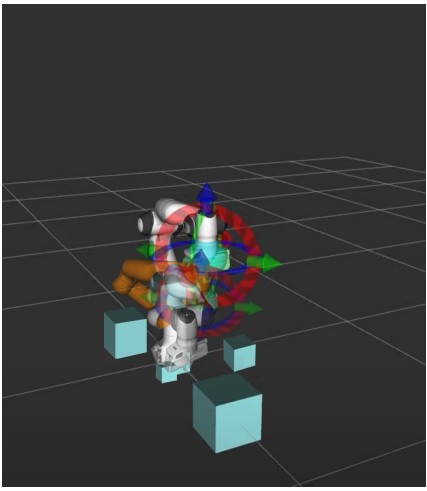

Figure 8: 7DoF Robotic Arm in a training environment with randomly sampled obstacles (those cyan cubes).

## E   MORE RELATED WORK

Refer to the suggestions of reviewers, we summarize the main difference of EAT-C with the suggested reference in this section.

The sub-goal generation in (Pertsch et al., 2020) follows a top-down and coarse-to-fine manner. However, they need to search for each sub-goal in the tree from many possible candidates, **which is expensive and requires a search tree** (hierarchical partition of the whole sub-goal space) much larger than our sub-goal tree (see Eq. 2). On the contrary, EAT-C learns a planning policy to **directly generate su-bgoals** and we do not need to build the search tree covering the whole sub-goal space. Another major difference is that **they study a planning-only method while we study a mutual learning strategy** between planning and RL to improve both planning and RL policies.

Zhang et al. (2020) trains a high-level policy to find the shortest path of sub-goals in a trained adjacency space. However, the distance between any two points in the adjacency space is expected to reflect the time cost of the agent navigating between the two points in the environment, which **can be very challenging or even infeasible to achieve in many tasks** (If we have such an adjacency space, both planning and RL can have dense feedback and simple supervised learning should work). In contrast, EAT-C trains a planner to directly generate a min-cost path of sub-goals through an easy-to-hard curriculum (fewer sub-goals interpolated at first), which provides an easier and more efficient solution **without requiring learning an adjacency space**. Moreover, the data used to train the planner in EAT-C are more informative than (Zhang et al., 2020) and cover multi-granularity since they are collected from RL when completing the bottom-up sub-task curriculum.

In (Dayan & Hinton, 1993), the high-level managers set a sequence of subgoals in the environment partitioned by Euclidean distance, which does not consider the obstacles or the RL agent capability. Hence, **there is no mutual training between high-level (planner) and low-level (controller) managers in** (**Dayan & Hinton, 1993**). On the contrary, the planner in EAT-C is jointly trained with the RL agent to produce a min-cost path of sub-goals for RL, which results in a more efficient curriculum of sub-tasks to train the RL agent.

Zhang et al. (2020) and Schmidhuber & Wahnsiedler (1993) plan sub-goals sequentially from the starting state to the goal state, which might be inefficient in complex tasks (requiring expensive search in a large space) and **cannot produce the easy-to-hard curricula on a sub-goal tree as in EAT-C**. In contrast, we train a planner to recursively produce coarse-to-fine sub-goal trajectories between the starting and goal states, which naturally provide an easy-to-hard curriculum for every component.

