# OpenReview forum: "EAT-C: Environment-Adversarial sub-Task Curriculum for Efficient Reinforcement Learning"
_ICLR.cc/2022/Conference — ICLR 2022 Submitted_

### Official Review · Reviewer_Zhvq · 2021-10-23

**Correctness:** 4
**Technical Novelty And Significance:** 2
**Empirical Novelty And Significance:** 2
**Recommendation:** 6
**Confidence:** 4

**Main Review:**

In general, I enjoyed reading the paper. It is clearly written and does not try to oversell its contribution - the combination of a recent hierarchical goal-based RL approach and adversarial environment generation. My main questions/critiques revolve around the experimental section. I can imagine that some of them may have been already addressed in the appendix that has been mentioned in the main paper but seems to be missing in the current revision:

* As I already mentioned, I think that the authors are in general objective in their assessments of the novelty of their method.
Only one statement on page 2 caught my eye: "Unlike previous methods, training these two policies does not require external supervision or prior knowledge, we instead collect the time costs and rewards of the RL agent on previous sub-tasks to train them towards generating better curricula adaptive to RL progress, which will be used in the next episode." I think that there already exist hierarchical RL approaches that do not need additional feedback or priors in addition to the MDP specification, such as e.g. [1].
* The authors compared their method to curriculum RL methods such as ALP-GMM. For these curriculum RL algorithms, an appropriately defined parameterization of the MDPs is crucial to obtain good learning performance. Unfortunately, details regarding this parameterization were not included in the main paper. For the curriculum RL methods, I would expect the parameterization to include both the parameters that the EG in EAT-C can control as well as the goal state to be reached. In this setting, to comparison is the most fair and consequently, the benefit of the particular structure of the proposed approach becomes the deciding factor in the performance difference. Could the authors clarify on the environment parameterization for the curriculum RL methods? If it differs from the aforementioned one, an additional investigation with this particular parameterization would clearly improve the paper.
* Another point of confusion is the very poor performance of the hierarchical RL baseline, which is unexpected given the clear benefit that a reader would expected from hierarchical RL in long horizon tasks. Although I can imagine that this poor performance may be caused by the randomization over environment configurations, the authors should consider providing additional evidence for this hypothesis. This could be done by:
	* Evaluating the hierarchical RL method also in the non-randomized environment used for the ablations of EAT-C.
	* Evaluating the ablation of EAT-C without the EG in randomized environments.
	* Clarify details regarding the observability of the environment configuration (see next point).
* Is the configuration of the environment hidden from the RL agent? Or is the agent aware of e.g. obstacle/object positions via observations? I am asking because compared to the experiments by Yamada et al in the 2D pusher task, much more samples are required to reach acceptable performance. So it seems that the generalization over environment configurations induces large challenges. Hence it would be important to carefully specify how the environment randomization is perceived by the agent.
* A more "realistic" experiment could further improve the paper. Such an experiment could be e.g. the 7-DoF planning environment with obstacles done in the sub-goal trees paper by Jurgenson et al.
* Could the EG in theory create infeasible tasks? Does this need to be prevented by the appropriate parameterization of the environment/EG actions?
* In algorithm 2: Could the REACH procedure possibly run into an infinite recursion with a very poor planning policy $\pi_p$ that always predicts infeasible goals? If so, could this be avoided with some naive pre-training? If this infinite recursion cannot happen, what prevents the possibly infinite recursion?

[1] Kulkarni, Tejas D., et al. "Hierarchical deep reinforcement learning: Integrating temporal abstraction and intrinsic motivation." Advances in neural information processing systems 29 (2016): 3675-3683.

**Summary Of The Paper:**

The paper proposes an RL method targeted at long-horizon tasks in environments with perturbed dynamics or configurations. By combining a recently proposed goal-based RL method with an adversarial approach to domain randomization, the authors obtain a method that allows them to learn robust policies in sparse long-horizon settings. The method is compared with different baselines form hierarchical- or curriculum RL, demonstrating promising performance.

**Summary Of The Review:**

Despite its rather incremental nature, I like the overall paper. Mainly the questions w.r.t. experimental details and the smaller scale evaluation tasks prevent me from putting it above the acceptance threshold. If some of my issues/questions are addressed, I am happy to improve my score.

---

> ### Author Response · Authors · 2021-11-23
> **Response to reviewer Zhvq (PART 2)**
>
> ### Q6: ‘’Another point of confusion is the very poor performance of the hierarchical RL baseline...’’
> - As you suggest, **we run a new group of experiments for the ablation study and report the results in the table below**. In the training environment (non-random and fixed) used in our original ablation study of Fig. 2(b), Hierarchical RL (HRL) does improve the performance of the default RL algorithm (SAC) on long-horizon tasks, i.e., 39.62 (SAC) vs. 68.27 (HRL). **However, when tested on random and unseen environments, HRL generalizes much poorer than EAT-C**.
> |Test Setting | Multiple New Random Environments | Training Environment (non-random) |
> |:--|:--|:--|
> | EAT-C | 80.24$\pm$12.25 | 92.04$\pm$6.49 |
> | EAT-C (remove EG) | 42.23$\pm$10.34 | 85.47$\pm$9.12 |
> | EAT-C (remove planner) | 27.58 $\pm$ 14.67 | 46.02$\pm$10.37 |
> | SAC | 20.83 $\pm$ 7.24 | 39.62 $\pm$ 12.25|
> | Hierarchical RL | 22.04 $\pm$ 10.44| 68.27 $\pm$ 6.99|

---

> ### Author Response · Authors · 2021-11-23
> **Response to reviewer Zhvq (PART 1)**
>
> Thank you for your time and suggestions! As you suggested, we have conducted experiments on the 7DoF robotic arm and reported the results in Appendix D.3 of the updated paper. We have also added more discussion of more related work. In the following, we will address your main concerns:
>
> ### Q1: “ I think that there already exist hierarchical RL approaches that do not need additional feedback or priors in addition to the MDP specification, such as e.g. [1]”
>
> - We agree that [1] does not require priors except the MDP specification. We have corrected our statement in the paper. However, **it is worth noting the following important difference and advantages of EAT-C compared to [1]**:
>
>    **(1)** The meta-controller in **[1] predicts sub-goals sequentially** from the starting state to the goal state, which is challenging and inefficient. In contrast, the planner in **EAT-C builds the sub-goal in a coarse-to-fine manner** and the top-down construction forms an easy-to-hard curriculum improving the training efficiency of the planner.
>
>    **(2)** The mutual training scheme and curricula in **EAT-C provide dense feedback** to train the planner and RL, whereas **[1] utilizes the sparse environment reward** to train the meta controller, which is less efficient on long-horizon tasks.
>
> ### Q2:  “For these curriculum RL algorithms, an appropriately defined parameterization of the MDPs is crucial to obtain good learning performance...I would expect the parameterization to include both the parameters that the EG in EAT-C can control as well as the goal state to be reached.”
>
> - **Our comparison is fair**: in all experiments, curriculum RL algorithms such as ALP-GMM **control exactly the same environment parameters as EG in EAT-C. They also control the sub-goals to be reached** when selecting/generating tasks. Appendix A in the new version elaborates on the details of environment parameterization.
>
> ### Q3: ‘’much more samples are required to reach acceptable performance.’’
> - First, the 2D-pusher tasks in our experiments **require both navigation and control**, so they are **very different from the control-only tasks in Yamada et al. and thus more challenging**: The robot in our tasks needs to first navigate to a state close to the object and then push it to the goal state. In contrast, the endpoint of the robotic arm in Yamada et al.’s experiments is fixed and does not need to move.
> - Second, **our test setting is much more challenging than the one used in Yamada et al.** since we evaluate the RL policy on **multiple random and unseen environments** different from the one used for training.
>
> ### Q4: ‘’Could the EG in theory create infeasible tasks? ‘’
> - We **always conduct a connectivity test** to the generated environments so the accepted **environments used for training are all feasible**. It is worth noting that EG is always trained on feasible environments in EAT-C so it tends to generate feasible ones. This was verified in our experiments: we observed that most of its generated environments can pass the connectivity test.
>
> ### Q5: ‘’In algorithm 2: Could the REACH procedure possibly run into an infinite recursion with a very poor planning policy $\pi_p$  that always predicts infeasible goals?’’
> - In earlier stages when $\pi_p$ and the RL agent are not well trained, $\pi_p$ may generate hard sub-goals and we need to recursively run REACH several times until the sub-tasks are sufficiently easy for the RL agent to solve, e.g., in the extreme case, a sub-goal near the agent such that one step completes the sub-task.
> - However, after a little training on the sub-task curriculum, $\pi_p$ is trained to generate a minimum-cost path for the RL agent, and the capability of the RL agent to finish the sub-tasks is also improved, **so the chance to trigger the recursion of REACH becomes very small**.
> - In practice, we can **set maximal times of recursively calling REACH and skip the unfinished sub-task** to avoid infinite recursion. In our experiments, we always observe that the **time cost that the agent needs to complete each sub-task decreases significantly** during training, indicating that $\pi_p$ does not propose infeasible goals. This can be verified by Figure 7 of the Appendix in the new version:  we plot how the average time-cost per sub-task at layer-3 of the sub-task tree changes during training.
>
> [1] Kulkarni, Tejas D., et al. "Hierarchical deep reinforcement learning: Integrating temporal abstraction and intrinsic motivation." Advances in neural information processing systems 29 (2016): 3675-3683.

---

> > ### Comment · Reviewer_Zhvq · 2021-11-23
> > **Thank you for your response**
> >
> > I appreciate the authors' efforts to improve the paper via additional experiments and clarifications. Unfortunately, I cannot raise my score due to additional concerns that have been raised by the authors' answer:
> >
> > * In their response, the authors mention that the baseline algorithms also control the sub-goals to be reached. As detailed in the new appendix A.3, this seems to be not the case at least for ALP-GMM, which can only control the parameters of the obstacles, but not the goals/objects. However, it would be easy to allow ALP-GMM to also control the goal position, object position and even the initial state of the robot. Further, it reads like ALP-GMM is implemented in a different way than proposed by Portelas et al., as the authors write that „each baseline generates an environment of obstacles by uniformly sampling the four parameters defining each obstacle from the corresponding ranges […] and can take a mutation step one time to change the lower and upper bounds of each parameter’s range“. In regular ALP-GMM, a  Gaussian mixture model over the (promising) parameter space is learned and directly sampled as opposed to uninformed sampling + a mutation step. Such details need to be presented with more clarity and should ideally be accompanied by code.
> >
> > * The additional experiments should definitely be added to the main text. Table 4 is much more insightful than Figure 2b and Figure 8 and Table 5 are also much more interesting than the rather small scale discrete space tasks. Those results can e.g. replace Section 5.3, which in my opinion is better suited for an extended presentation in an appendix than the new experimental results.
> >
> > * Finally, the authors mentioned a connectivity check for the adversarial environment generator that I was not aware of after first reading the paper. I also cannot find such a step in Algorithm 3. Given that such a connectivity check requires additional knowledge (as checking feasibility may not be trivial for all environments), the authors should discuss this additional structure for the individual environments and potentially do an ablation, e.g. extending Table 4.
> >
> > To summarize, I think that the authors have a solid approach and experimental evaluation. However, the current presentation - in my opinion - needs to be further improved and ideally, the code should be made publicly available such that it is easier to resolve detailed questions regarding the method.

---

> > > ### Author Response · Authors · 2021-11-23
> > > **Thank you for the quick response! But your new concerns are INCORRECT!**
> > >
> > > Thank you for your quick response (within 2 hours after the new version was uploaded) and for agreeing that we have **"a solid approach and experimental evaluation"**! We also appreciate that you acknowledged that the two groups of new experiments are more interesting: the improvement achieved by EAT-C on them is non-trivial. However, with all due respect, **the rest comments (the first and third bullet point) in your new response are INCORRECT**, partially due to our inaccurate statement (**we do apologize!**) about “Uniform sampling + a mutation step”, which is only for POET, not for ALP-GMM (we use its original algorithm without any change). We will make these points very clear in the following.
> > >
> > > ```
> > > Q1: ALP-GMM "which can only control the parameters of the obstacles, but not the goals/objects. However, it would be easy to allow ALP-GMM to also control the goal position, object position and even the initial state of the robot.", "it reads like ALP-GMM is implemented in a different way than proposed by Portelas et al.,"
> > > ```
> > >
> > > - **We use exactly the same ALP-GMM algorithm proposed by Portelas et al. and the original implementation with NO CHANGE**. ALP-GMM can control all the obstacles (2D-pusher) and the objects (discrete space tasks), whose positions and sizes are all included in the parameter $p$ in Algorithm 1 of the ALP-GMM paper [Portelas et al.].
> > > - **No method in this paper can control the initial states $s_0$ and goal positions $g$: every method is assigned with RANDOM $(s_0,g)$ pairs they CANNOT CONTROL**. This better matches the practical scenarios of online RL. Given a random $(s_0,g)$, EAT-C, SGT, and planning methods only have the freedom to interpolate sub-goals between $(s_0,g)$. ALP-GMM cannot interpolate sub-goals because it does not have any sub-goal planning in its algorithm.
> > > - **"Uniform sampling + a mutation step" holds ONLY for POET, NOT for ALP-GMM**. We will correct this inaccurate statement. We never change the original ALP-GMM algorithm that samples high-progress tasks from a learned GMM.
> > > - In summary, all evaluated methods including ALP-GMM and EAT-C CAN fully control all obstacles (2D-pusher) and objects (discrete space tasks) but CANNOT control the goal position and initial states (i.e., the assigned tasks). Hence, **the comparison is FAIR and ACCURATE**.
> > >
> > > ```
> > > Q2: "Given that such a connectivity check requires additional knowledge (as checking feasibility may not be trivial for all environments)"
> > > ```
> > >
> > > - The connectivity test aims at only sending solvable environments to different methods for training. **It does not change or affect the training process and it is FAIR to all methods**. Without the test, every method will receive a similar amount of unsolvable tasks during training and they will all fail (exceeding the maximum time limit) on these tasks because no policy can solve an unsolvable task. Therefore, removing the connectivity test brings almost the same degradation on the training efficiency for every method.
> > > - In EAT-C, EG is trained on solvable environments to directly generate bounded modifications without connectivity checking. We have reported the time-cost that the RL agent needs to complete sub-tasks in Fig.7, **the results demonstrate that EG is not making the environment unsolvable**.
> > >
> > > ```
> > > Q3: "The additional experiments should definitely be added to the main text"
> > > ```
> > >
> > > - We will add the new experiments to the main text. These experiments demonstrate that **EAT-C consistently outperforms a broad range of strong baselines** (curriculum RL, hierarchical RL, RL with environment-changing or planning-only) **across a diverse set of tasks** (navigation, complicated robotic arm control, navigation + control, compositional tasks). Moreover, as shown in Fig. 2(a) and Table 4, it significantly improves RL’s generalization and robustness on random and unseen environments during inference/deployment, e.g., **EAT-C outperforms the BEST baseline in Fig.2(a) by >20% and improves hierarchical RL by nearly 60% on the success rate!**
> > >
> > > Given the above answers and the "solid approach and experimental evaluation" with strengthened new experiments, do you still have concerns about the fair comparison and ALP-GMM implementation (please let us know)？Would you mind reconsidering your rating? Thanks！

---

> > > > ### Comment · Reviewer_Zhvq · 2021-11-24
> > > > **I still disagree w.r.t. baseline evaluations**
> > > >
> > > > I thank the authors for clarifying my remaining concerns especially regarding the implementation of the ALP-GMM baseline.
> > > >
> > > > However, I still think that ALP-GMM should be given control over some form of combination of goal-, object- and initial state. I understand that this may not seem fair to the authors, as their method (EAT-C) does not explicitly have control over these parameters of the environment. There are, however, two important arguments why this is still an important experiment to conduct:
> > > >
> > > > * From a practitioner's view, we are able to control the simulation as we wish and hence we may just grant control over goal-, object- and initial state in addition to the obstacle parameters. It is hence interesting to see whether EAT-C performs just as good to such an orthogonal approach. Note that I don't want to necessarily see EAT-C performing better, as it uses a different set of assumptions. However, it would be highly insightful to have this comparison and discussion in a paper.
> > > > * From a scientific view, the performance of ALP-GMM with control over the goal/object/initial state can shed light on the differences that arise from how structural assumptions are exploited in RL. In curriculum RL (i.e. ALP-GMM), we exploit the knowledge about environment parameters that make learning easier. I argue that Hierarchical RL also exploits similar concepts, but in a different way. More precisely, HRL assumes that you can e.g. decompose a "long" task into a sequence of "smaller" tasks and that each smaller task is easier to solve than the long task. Further, you assume that you can easily combine the solution to the small tasks to obtain a solution for the long tasks. On an abstract level, this leads to the generation of a curriculum of intermediate tasks. While I totally agree that these assumptions and the resulting approach (EAT-C) are valid, it would still be interesting to compare the two different ways of exploiting assumptions about the task structure. This is why I am so strongly insisting on this additional experiment. I truly believe that it would greatly improve the paper.
> > > >
> > > > Given that the authors say that my statement regarding the non-described connectivity check was incorrect, can they please refer me to the position in the first revision of the submitted document, where this has been stated and discussed? If I have overlooked it during my first review, I apologize for the wrong statement.

---

> > > > > ### Author Response · Authors · 2021-11-27
> > > > > **We add the new experiments you requested! Our previous setting is more practical and is fair to all baselines. (Part2/2)**
> > > > >
> > > > > ```
> > > > > Q2: “However, I still think that ALP-GMM should be given control over some form of combination of goal-, object- and initial state”, ”From a practitioner's view, we are able to control the simulation as we wish and hence we may just grant control over goal-, object- and initial state in addition to the obstacle parameters.”
> > > > > ```
> > > > >
> > > > > - **Our previous evaluation setting is fair to all methods. It is also more practical** because the targeted tasks and their initial/goal states may have already been defined by users, so the curriculum can only decompose each predefined task into a sequence of sub-tasks that can be learned more efficiently. For example, if a human user needs to train a robot to finish some kitchen tasks whose initial and goal states are predefined according to the user's habit, the curriculum should not change the human orders.
> > > > > - **That being said, we totally understand your intuition of the new evaluation setting**, in which the curriculum has all the control. We would like to argue that **this setting is more restricted in practice** than our setting since it requires controlling more attributes (e.g., the initial states, final goals, and assigned tasks). Moreover, these curriculum-selected tasks are not randomly sampled from some well-defined distribution, so it is hard to analyze or qualify the generalization of the learned policy. Hence, a curriculum that controls everything of training tasks can bring severe bias, distribution drift over time, and can lead to poor generalization or catastrophic forgetting. In contrast, we keep all the assigned tasks intact in EAT-C and they can be randomly sampled from a distribution. The curriculum in EAT-C only changes the sub-tasks but all the sub-tasks are designed to accomplish the original tasks.
> > > > >
> > > > > ```
> > > > > Q3: “I argue that Hierarchical RL also exploits similar concepts, but in a different way.”
> > > > > ```
> > > > >
> > > > > We appreciate your discussion and comparison between Hierarchical RL and ALP-GMM. We agree that they adopt two different types of curriculum. We would like to bring your attention to **the difference between Hierarchical RL (HRL) and EAT-C**, though we are inspired by HRL’s idea of decomposing a hard task into easier sub-tasks:
> > > > >
> > > > > - **EAT-C does not need to build a hierarchical partition of the whole action/sub-task space as HRL**, which is very expensive. It does not require pre-defined sub-tasks. Instead, our subtask tree is hierarchical planning for a given task and it is directly generated by our planning policy rather than by a search algorithm on a hierarchy;
> > > > > Although the idea of decomposing a hard task into easier sub-tasks in EAT-C is inspired by HRL, the **adversarial environment modification for each sub-task has not been explored in HRL** but is essential to improving the training efficiency, because the sub-tasks with the same environments might be redundant to learning and some might be too easy for the RL agent.
> > > > > - **The mutual training among the planner (higher-level policy), the RL agent (lower-level policy), and the environment generator in EAT-C** has not been explored in HRL but it is important to generate dense rewards and an easy-to-hard curriculum to train every policy more efficiently, without external supervision.

---

> > > > > > ### Comment · Reviewer_Zhvq · 2021-11-29
> > > > > > **Thank you for the additional experiments**
> > > > > >
> > > > > > I am happy that the authors performed additional experiments and the explanations for the encountered problems seems convincing to me. I hence increased my score to a weak accept. The reason for a weak accept are the many changes that are required to bring the additional results into an insightful ordering for the reader. However, I believe that with these changes, the paper can be an insightful contribution to the RL community.
> > > > > >
> > > > > > I have one final point of disagreement (which the authors however do not need to resolve): HRL methods do not necessarily need to create a partition of the action/sub-task space in an expensive manner. A "goal-setting" upper-level policy does not incur overhead except for training this upper-level policy (which EAT-C also needs to do to). Note that I do not want to argue for such an approach over the proposed method, but I just want to point out that a bit more care should be taken with such strong statements.

---

> > > > > ### Author Response · Authors · 2021-11-27
> > > > > **We add the new experiments you requested! Our previous setting is more practical and is fair to all baselines. (Part1/2)**
> > > > >
> > > > > ```
> > > > > Q1: “the performance of ALP-GMM with control over the goal/object/initial state can shed light on the differences that arise from how structural assumptions are exploited in RL”, “This is why I am so strongly insisting on this additional experiment. I truly believe that it would greatly improve the paper.”
> > > > > ```
> > > > >
> > > > > - **As you suggested, we add new experiments of ALP-GMM that can control everything (initial and goal states, obstacles, object)** in 2D-pusher (we are still running experiments for ‘’discrete space tasks’’ and will post the results when we get them). Specifically, ALP-GMM can control the location and size of the obstacles, the initial/goal state, and the location of the object by sampling from the learned GMM.
> > > > > - **We carefully tuned the hyperparameters of ALP-GMM in the new setting by grid search** and chose the one achieving the best performance. Specifically, for the hyperparameters in Algorithm 1 of ALP-GMM paper (https://arxiv.org/pdf/1910.07224.pdf): (1) we tune the probability of random sampling $p_{rnd}\in [0.05,0.5]$ and the best one selected is $p_{rnd}=0.25$; (2) we tune $k_{max}\in\{4,6,8,10,12\}$ and the number of Gaussians is adapted online by fitting multiple GMMs (here having from $4$ to $k_{max}=12$ Gaussians); (3) we tune the fitting rate $N\in\{200,250\}$ and the best one is $N=250$, the same as ALP-GMM paper.
> > > > > - **We report the performance of the new “ALP-GMM (control all)”**on test tasks over the course of training in the table below. We also include our previous results of EAT-C and ALP-GMM for comparison. Note these two cannot control the original tasks, i.e., the initial state $s_0$ and the final goal state $g$.
> > > > > - **The new experiments show that (1) EAT-C with partial control still significantly outperforms ALP-GMM (control all), i.e., $81.45$ vs. $61.05$; and (2) ALP-GMM (control all) can surpass ALP-GMM with partial control in the middle stage of training ($54.52$ vs. $48.15$ at 10.0 million env. steps) but its final performance is worse ($61.05$ vs. $66.21$).** This can be explained by our analysis in the response to your first comment: curriculum having total control over the assigned tasks can introduce bias and distribution drift over the course of online RL. **Therefore, randomly drawing the assigned tasks but building a curriculum within each task, which is how EAT-C works in our evaluation setting, is more robust.**
> > > > >
> > > > > | Environment Steps | 5.0 million steps |  10.0 million steps | 15.0 million steps | 20.0 million steps |
> > > > > |:--|:--|:--|:--|:--|
> > > > > | ALP-GMM | 18.33 $\pm$ 14.25 | 48.15 $\pm$ 11.34 | 62.35 $\pm$ 8.47 | 66.21 $\pm$ 9.35 |
> > > > > | ALP-GMM (control all) | 34.28 $\pm$ 12.14 | 54.52 $\pm$ 14.33| 58.67 $\pm$ 10.54| 61.05 $\pm$ 12.45 |
> > > > > | EAT-C | 39.76 $\pm$ 13.46 | 69.45 $\pm$ 17.42 | 78.15 $\pm$  13.66 | 81.45 $\pm$ 11.35 |

---

### Official Review · Reviewer_3c96 · 2021-11-02

**Correctness:** 3
**Technical Novelty And Significance:** 3
**Empirical Novelty And Significance:** 3
**Recommendation:** 6
**Confidence:** 3

**Main Review:**

======Strengths======

1. The main idea of this approach is easy to follow and seems intuitive. The approach also shows promising results in the experiments. Though evaluated in simple tasks, applications in harder environments and tasks might be plausible if the subgoal decomposing and the environment generation could be adapted to more difficult environments and tasks.

2. The figures and the qualitative examples are very helpful for readers to understand the core idea as well as how the auto-curriculum really works in the experiments.

======Weaknesses======

1. While the main idea of the algorithm is clearly presented, implementation details are missing. E.g., I could not find information about how the environment generation was implemented (e.g., action space). A more detailed description and/or code would be very helpful.

2. The performance of the algorithm may heavily depend on the tree structure generation approach. The current work adopts a specific kind of approach, which may not work for other kinds of tasks. It would be good to discuss the limit/generalizability of the overall algorithm.

3. The conclusion section is missing. It would be good to summarize the main contributions and findings at the end of the paper.

Some suggested references:

[1] Kaelbling, L., & Lozano-Perez, T. (2011). Hierarchical task and motion planning in the now. In Ieee international conference on
robotics and automation. (Classic work on task and motion planning, a well-known type of hierarchical planning.)

[2] Shu, T., Xiong, C., & Socher, R. (2018). Hierarchical and interpretable skill acquisition in multi-task reinforcement learning. In ICLR. (Human-designed curriculum for learning the composition of subgoals via a tree structure.)

[3] Baker, B., Kanitscheider, I., Markov, T., Wu, Y., Powell, G., McGrew, B., & Mordatch, I. (2019). Emergent tool use from multi-agent autocurricula. In ICLR. (Autocurricula emerged from adversarial multi-agent RL training.)


**Summary Of The Paper:**

This paper introduces an auto-curriculum generation approach to train better low-level policies conditioned on the subgoals generated by a path-planning high-level policy. The core idea is to use a planning policy to decompose a task into subgoals via a tree structure (multi-level subgoal decomposition) and use an environment generation policy to adversarially make the environment harder for the subgoal policy to perform. Thanks to the tree structure, the adversarial training can follow a bottom-up process, i.e., in an easy-to-hard order. The evaluation on a 2D object using task and a 2D grid world environment shows the effectiveness of this approach.

**Summary Of The Review:**

The proposed algorithm presents an interesting and novel way to adversarially generate a curriculum for training low-level RL policies. The algorithm seems general to some extent and performed well in two tasks. However, the writing needs improvement (a lack of details, missing conclusion, etc.). There should also be a discussion on the limit of the current algorithm, which relies on the performance of a specific plan-planning method that provides the true structure. I am willing to raise my score if these problems are addressed in the revision.

---

> ### Author Response · Authors · 2021-11-23
> **Response to reviewer 3c96**
>
> Thank you for your time and suggestions! We have included the suggested reference in the new version. We address your main concerns as below:
>
> ### Q1:’’how the environment generation was implemented (e.g., action space)’’
>
> - **We have uploaded a new version of the paper that contains the details of environment generation in Appendix A**. For EAT-C in the 2D-pusher experiment, we represent the environment by $\theta$ that comprises multiple tuples in the form of $(x_i,y_i,w_i,h_i)_n$ , each defining a rectangle obstacle with location coordinates $(x_i, y_i)$ and size (width and height) $(w_i,h_i)$. The environment generator (EG) can modify each obstacle in the environment by changing its location and size, where the change is $b_t = (\Delta x_i, \Delta y_i, \Delta w_i, \Delta h_i)$, i.e., an action taken by EG. We also constrain the changes in $b_t$ from being too large by thresholding, similar to adversarial attacks.
>
> ### Q2: ‘’ The current work adopts a specific kind of approach, which may not work for other kinds of tasks’’
>
> - EAT-C is more than a specific kind of approach: **it provides a principal framework to efficiently train an RL agent** on a more informative and adaptive curriculum of tasks automatically generated by the planner and environment generator (EG). And training the planner and EG does not make the problem harder due to the mutual training scheme and the auto-generated curricula for them.
> - The three components in EAT-C, i.e., **the RL agent, the planner, and EG, are generally defined and can be trained using any existing off-policy RL algorithm** (as mentioned in Algorithm 2 and the last paragraph on page 6). This is because: (1) the planner is general since the distance metric for planning is the time cost of the RL agent and thus can be applied for any environments and tasks; (2) EG is formulated as another general RL problem so both the RL agent and EG are general RL agents.
>
> ### Q3: “The conclusion section is missing.”
>
> - We have added a conclusion section in our new version, i.e.:
> - We propose a mutual learning and auto-curriculum framework “EAT-C” to improve the efficiency of RL on long-horizon tasks as well as its generalization and robustness to new environments. EAT-C trains a planner to decompose a hard task into coarse-to-fine sequences of sub-tasks providing an easy-to-hard curriculum to train an RL agent, while an adversarial environment generator modifies these sub-tasks to be diverse and more informative to learn. The three policies are trained with data collected by each other. On two types of tasks, EAT-C outperforms a diverse set of baselines, e.g., curriculum-based RL, hierarchical RL, and planning-based methods. It has the potential to be applied to more complicated tasks with dynamic environments or visual inputs such as games, which will be covered in our future works.

---

### Official Review · Reviewer_o38w · 2021-11-03

**Correctness:** 3
**Technical Novelty And Significance:** 2
**Empirical Novelty And Significance:** 2
**Recommendation:** 5
**Confidence:** 4

**Main Review:**

The biggest strength of this paper is in its combination of two important and productive lines of research from HRL and environment design.  The combination of the two is relatively natural because they solve different aspects of sparse reward problem, and to the extent there are synergies between approaches, this is a productive direction of research.

However, it is not clear if what is achieved by this approach could be achieved by running existing HRL techniques in the context of adversarial environment design. The one thing that makes this papers approach distinct from this direct combination is that the adversarial design is done independently for every sub-goal.  However, it is not clear that this distinction is necessary for the approach to work.  In fact, the ablation study in Figure2 (b) leads me to believe that the adversarial environment design component is relatively unimportant, so I wouldn't be surprised if the somewhat marginal benefits could also be achieved by adversarial environment design separate from the hierarchy.  If this is the case, then it is difficult for me to see the benefits of combining the approaches, given the added complexity, added dependencies, and marginal benefits.

In terms of the benefits, the empirical results are also inconclusive.  One concern is that SAC by itself beats some of the baselines, which makes it seem like the baselines may not be configured properly.  In particular, POET has environment-dependent parameters which need to be tuned, so it is not obvious to me how this baseline works, or how it was ensured that these parameters were set correctly.  Even without these concerns, it seems that most of the results do not seem to be completely statistically significant.

The next major concern is that it seems that it would be easy for many of the components to fall into undesirable equilibrium.  For instance, the environment adversary could make unsolvable environments, and it seems like the action space of the environment adversary had to be restricted to avoid this.  Also if the agent does not have some capability, and the planner would not ask the agent to solve those goals since it is cooperative, and thus the agent would not gain experience for those goals, which could also halt progress.

Finally, it is unclear to me if the hierarchy of subgoals being mutually trained with the RL agent is novel.  It appears similar to a few of the following and there should at least be a discussion of how these differ, if that is to be taken as a contribution:
"Long-Horizon Visual Planning with Goal-Conditioned Hierarchical Predictors" Pertsch K and Rybkin O, et. al.
"Generating Adjacency-Constrained Subgoals in Hierarchical Reinforcement Learning" Zhang, T and Guo, S et.al
"Feudal reinforcement learning" Dayan, P and Hinton, G et.al
"Planning simple trajectories using neural subgoal generators" Schmidhuber, J and Wahnsiedler, R et.al

Minor point:
* The paper ends suddenly, I was expecting a conclusion section
* I didn't realize the legends where different for Figure 2(a) and Figure 2(b), it would be less confusing if the reuse of colors could be avoided
* The caption for Figure 2 blends into the text.
* All references for Figure 3 are before all references to Figure 2, but they are displayed in the other order
* I do not know what is meant by a "mutual boosting scheme", the term is used several times but does not seem to be defined.
* The second to last paragraph of page 3 there is a typo: "requires to collect"
* In the first paragraph of page 6, you first mention that the RL agent is trained on one rollout-per-layer rather than one rollout just for the leaves of the hierarchy.  This should be made clearer earlier, as it is a significantly different method with this change.
* In the second paragraph of page 6 there is the claim that the sub-goal curricula is "an imitation of human learning that repeated practice the same complicated task in different ways".  I do not believe this claim, though I am not sure I understand it, but I do not think it is necessary to understand the method.
* There is also a claim at the end of that paragraph that they adversary generates diversity.  I do not know why that would be true, it seems that the adversary could easily fixate on a particular type of modification.
*The generic idea that environment design techniques make environments more reward sparse (for instance, at the end of the first paragraph of page 1), is true of fully-adversarial environment design techniques, but it is not true for techniques like:
"Intrinsic Motivation and Automatic Curricula via Asymmetric Self-Play" Sukhbaatar, S et al.
"Emergent Complexity and Zero-shot Transfer via Unsupervised Environment Design" Dennis, M, et al.
both of which are designed to densify the reward structure by designing hard but solvable levels, and would likely serve as better baselines.  In particular, the method used for the environment designer in this work appears to correspond to the "maximin adversary" in the second paper, which was used as a baseline.

**Summary Of The Paper:**

This paper proposes a method for generating curriculum for goal-conditioned RL policies based on having one agent hierarchically generate sub-goals, and have the target agent train from rollouts on each level of this hierarchy, while a third agent adversarially modifies the environment to improve robustness and transfer.  The main contributions are introducing a mutual training regime between the path planer and the RL agent, and introducing adversarial environment design to training in each of the sub goals.

**Summary Of The Review:**

I will be weakly recommending rejection since it is unclear to what extent the method works because of the integration of HRL and adversarial environment design, and to what extent the same results could be achieved through the simpler direct combination of the existing approaches in both fields.  In addition, it seems that the baselines could be made stronger, and I am not convinced by the empirical results.

---

> ### Author Response · Authors · 2021-11-23
> **Response to reviewer o38w (PART 4)**
>
> ### Q9: ‘’The generic idea that environment design techniques make environments more reward sparse’’
> - Adversarial environment design applied to the sub-task curriculum will make the sub-tasks **more diverse and sufficiently challenging but will NOT make the reward harmfully sparse, and here is why**:
> - **We apply EG to simple sub-tasks** that are optimized to be simple (via optimizing the planner) for the RL agent. The goal of EG is to avoid learning similar and easy sub-tasks repeatedly, which CANNOT provide informative feedback to RL even if the reward is dense.
> There are **several restrictions to avoid over-adversarial environments**:
>
>    **(1)** EG is trained to generate adversarial modifications for different environments so it is not finding the most adversarial perturbation for every environment but finding adversarial patterns generalized to different environments;
>
>    **(2)** Similar to $\epsilon$-ball constraints in adversarial attacks, we restrict EG not to modify each environment parameter too much.
>
> - **Our experiments provide direct evidence showing that EG improves the rewards rather than making it more sparse**:
>
>    **(1)** The time cost of the RL agent significantly reduces between episode = 3 to episode = 6 in Figure 4;
>
>    **(2)** In Figure 7 of the Appendix in the new version, the average time-cost per sub-task at layer-3 of the sub-task tree drastically decreases during training. The ablation studies, both the old one (Fig. 2(b), test on the same environment as training) and the new one (Q2 for Reviewer o38w, test on different unseen environments), show that EAT-C with EG outperforms EAT-C (remove EG) by a large margin and demonstrates the positive effects (and importance) of having EG in EAT-C.

---

> ### Author Response · Authors · 2021-11-23
> **Response to reviewer o38w (PART 3)**
>
> ### Q8: ‘’it is unclear to me if the hierarchy of subgoals being mutually trained with the RL agent is novel’’
> - Although it is not uncommon to have a hierarchy of subgoals in previous works, **we are the first to develop a mutual training scheme among the planner, RL, and EG**. This is the key contribution of EAT-C because it allows each component to overcome its own bottleneck by learning from the others and results in adaptive and efficient auto-curricula introduced in Section 4.1, so every component can learn from easy-to-hard tasks.
> - We have cited and discussed the suggested papers in our new version ([1] was covered in the last paragraph on page 3 of the original version). In summary, our main advantage and novelty is the mutual training scheme and the auto-curricula, which is the key to overcome several bottlenecks of previous works. We discuss the main difference between EAT-C and these methods in the following:
> The subgoal generation in [1] follows a top-down and coarse-to-fine manner. However, they need to search for each subgoal in the tree from many possible candidates, **which is expensive and requires a search tree** (hierarchical partition of the whole subgoal space) much larger than our sub-goal tree (see Eq. (2)). On the contrary, EAT-C learns a planning policy to **directly generate subgoals** (much faster than search!) and we do not need to build the search tree covering the whole subgoal space. Another major difference is that **they study a planning-only method while we study a mutual learning strategy** between planning and RL to improve both planning and RL policies.
> - [2] trains a high-level policy to find the shortest path of subgoals in a trained adjacency space. However, the distance between any two points in the adjacency space is expected to reflect the time cost of the agent navigating between the two points in the environment, which **can be very challenging or even infeasible to achieve in many tasks** (If we have such an adjacency space, both planning and RL can have dense feedback and simple supervised learning should work). In contrast, EAT-C trains a planner to directly generate a min-cost path of subgoals through an easy-to-hard curriculum (fewer subgoals interpolated at first), which provides an easier and more efficient solution **without requiring learning an adjacency space**. Moreover, the data used to train the planner in EAT-C are more informative than [2] and cover multi-granularity since they are collected from RL when completing the bottom-up sub-task curriculum.
> - In [3], the high-level managers set a sequence of subgoals in the environment partitioned by Euclidean distance, which does not consider the obstacles or the RL agent capability. Hence, **there is no mutual training between high-level (planner) and low-level (controller) managers in [3]**. On the contrary, the planner in EAT-C is jointly trained with the RL agent to produce a min-cost path of sub-goals for RL, which results in a more efficient curriculum of sub-tasks to train the RL agent.
> - [2] and [4] plan sub-goals sequentially from the starting state to the goal state, which might be inefficient in complex tasks (requiring expensive search in a large space) and **cannot produce the easy-to-hard curricula on a sub-goal tree as in EAT-C**. In contrast, we train a planner to recursively produce coarse-to-fine sub-goal trajectories between the starting and goal states, which naturally provide an easy-to-hard curriculum for every component.
>
> [1] Long-Horizon Visual Planning with Goal-Conditioned Hierarchical Predictors, Pertsch K and Rybkin O, et. al.
>
> [2] Generating Adjacency-Constrained Subgoals in Hierarchical Reinforcement Learning" Zhang, T. and Guo, S. et.al
>
> [3] Feudal reinforcement learning, Dayan, P., and Hinton
>
> [4] Planning simple trajectories using neural subgoal generators" Schmidhuber, J and Wahnsiedler, R et.al

---

> ### Author Response · Authors · 2021-11-23
> **Response to reviewer o38w (PART 2)**
>
> ### Q3: ‘’One concern is that SAC by itself beats some of the baselines, which makes it seem like the baselines may not be configured properly.’’
> - **This is wrong! No result in this paper shows that SAC by itself can beat any baseline**. The only result of SAC-only in this paper is in Figure 2(b), which shows that SAC performs much poorer than any variant of EAT-C. We did not report SAC’s performance in other plots/tables because it is a building block of every method evaluated in this paper and SAC-only produces much poorer results than any of the baselines in this paper. We adjusted the colors in Figure 2(b) to avoid misreading.
>
> ### Q4: ‘’In particular, POET has environment-dependent parameters which need to be tuned’
> - **The comparison to POET is fair: we use exactly the same representation of environments** in both EAT-C and POET and **carefully tuned all its hyperparameters to get its best performance**. We have not changed any step in the original POET algorithm, e.g., its perturbation of environments.
> A primary advantage of EAT-C compared to POET on environment modification is that we train an EG to automatically generate adversarial environments that are most informative to RL, while POET relies on random sampling and mutations, which can include many redundant or less informative environments. This can explain the better performance of EAT-C compared to POET in the experiments.
>
> ### Q5: ‘’the environment adversary could make unsolvable environments’’
> - We always conduct a connectivity test to the generated environments so the accepted ***environments used for training are all solvable***. We have the same test for other baselines. It is worth noting that the environment generator in EAT-C is always trained on solvable environments so it tends to generate solvable ones. This is verified in our experiments: we observed that most of its generated environments can pass the connectivity test.
>
> ### Q6: ‘’The next major concern is that it seems that it would be easy for many of the components to fall into undesirable equilibrium’’
> - **We have never observed any “undesirable equilibrium” in all the experiments so far**. It is unlikely to happen because every component in EAT-C starts from easy tasks for it at first before tackling more complicated ones, thanks to the top-down and bottom-up curricula and mutual boosting among them, which are our main contributions explained in Section 4.1.
>
> ### Q7: ‘’I do not know what is meant by a "mutual boosting scheme"’’
> - **As explained in the abstract, introduction, and Section 4.1**, the mutual boosting scheme refers to that the three policies, i.e., path-planner, RL agent, and environment generator (EG), generate tasks/data to train each other during the course of joint training. In this way, they can overcome the drawbacks of training each separately, and they do not rely on external supervision except the environment rewards.
> - In particular, the planner generates sub-tasks to train the RL agent with dense rewards, while EG adversarially modifies the environment in each sub-task to make those sub-tasks diverse and sufficiently challenging. Hence, **they together produce more informative tasks to train the RL agent**. On the other hand, the data collected during training the RL agent on those sub-tasks are then used to optimize the planner and EG: this is also important because it makes them (and their generated sub-tasks) adaptive to the RL agent’s need and the RL collected data provide dense rewards to train the planner.
> - Another advantage of the mutual boosting scheme is that **it naturally generates an easy-to-hard curriculum for every policy**:
>
>    **(1)** Top-down construction of the sub-goal tree produces an easy-to-hard curriculum for the path-planner since it only needs to interpolate fewer sub-goals at top layers;
>
>    **(2)** Bottom-up traversal of the sub-tasks on the tree to train the RL agent and EG is an easy-to-hard curriculum as well since they both receive dense rewards from bottom-layer sub-tasks at first before tackling longer-horizon tasks.

---

> > ### Comment · Reviewer_o38w · 2021-11-27
> > **Response to (PART 2)**
> >
> > > **This is wrong! No result in this paper shows that SAC by itself can beat any baseline**. The only result of SAC-only in this paper is in Figure 2(b), which shows that SAC performs much poorer than any variant of EAT-C. We did not report SAC’s performance in other plots/tables because it is a building block of every method evaluated in this paper and SAC-only produces much poorer results than any of the baselines in this paper. We adjusted the colors in Figure 2(b) to avoid misreading.
> > >
> >
> > Yes, Figure 2(b) is the only graph you mention SAC by itself.  But just next to it is Figure 2(a).  Both figures are test performance on the same environment "2D-Pusher". This appears to show that the performance of SAC is 20% above HRL and Value Disagreement in Fig. 2(a).  I now see that you mention in text that you have changed what the test set means for the ablation.  To avoid this confusion you should at least mention this in the caption, I could only deduce that this is what happened through cross-referencing section 5.2 and section 5.
> >
> > > **The comparison to POET is fair: we use exactly the same representation of environments** in both EAT-C and POET and **carefully tuned all its hyperparameters to get its best performance**
> > >
> >
> > Could you please point me to details about how this was done?  I can only find mention of the re-scaling factor for the reward (in appendix B), but there are many more parameter of POET.
> >
> > Just to list a few of these parameters:
> > * the "minimal criteria" parameters decide when an environment is too hard or too easy
> > * There is also the "reproduction eligibility condition" which decides when an environment is allowed to mutate to generate new children.
> > * There are limits on the number of new environments in any iteration "max_admitted"
> > * There are limits on the number of environments that can be active at a time "capacity".
> > * There is also a required novelty calculation which is unclear how you translate.
> > * In the original POET used a nearest-neighbor novelty bonus, which would need a parameter for the number of nearest neighbors.
> >
> > There are also hyper-parameters for ES:
> > * population size
> > * standard deviation of the noise
> > * the decay rate of the standard deviation parameter
> >
> > Of course there are the standard parameters for Adam.
> >
> > Given this large array of parameters, and their interactions, it is difficult to evaluate the extent to which POET was well-tuned.  Given that so many of these parameters are environment-specific, POET does not seem to have been used in this environment before, and even easier to-tune baselines are notoriously under-tuned in the field.  I find it difficult to accept the bolded claim that "carefully tuned all its hyperparameters to get its best performance" without an explanation of the procedure by which it was tuned or even a list of what the parameters ended up being.
> >
> > Tuning the hyperapameters well is a serious issue of reproducibility [1], these parameters dramatically impact performance, to the extent that they could completely change the result.  It has often been the case for prominent methods in the field that the results are all hyperapaemter tuning or low-level optimization details [2,3,4].  I cannot recommend the acceptance of a paper without these details reported, an assertion that it was done correctly is not enough.
> >
> > [1]Henderson, Peter, et al. "Deep reinforcement learning that matters." *Proceedings of the AAAI conference on artificial intelligence*. Vol. 32. No. 1. 2018.
> >
> > [2] Yu, Chao, Akash Velu, Eugene Vinitsky, Yu Wang, Alexandre Bayen, and Yi Wu. "The surprising effectiveness of mappo in cooperative, multi-agent games." *arXiv preprint arXiv:2103.01955* (2021).
> >
> > [3] Schroeder de Witt, C., Gupta, T., Makoviichuk, D., Makoviychuk, V., Torr, P. H., Sun, M., & Whiteson, S. (2020). Is Independent Learning All You Need in the StarCraft Multi-Agent Challenge?. *arXiv e-prints*, arXiv-2011.
> >
> > [4] Engstrom, L., Ilyas, A., Santurkar, S., Tsipras, D., Janoos, F., Rudolph, L., & Madry, A. (2020, January). Implementation Matters in Deep Policy Gradients: A Case Study on PPO and TRPO. In *International Conference on Learning Representations*
> >
> > > We always conduct a connectivity test to the generated environments so the accepted ***environments used for training are all solvable***
> > >
> >
> > This is a critical detail that is not adequately emphasized.  When mentioned to Reviewer Zhvq it was met with equal surprise, and neither of us have been pointed to it in the text so it seems to be a critical missing detail that significantly effects performance.
> >
> > This also greatly limits the applicability of the approach, as it is difficult to check that an environment is solvable without first having an approach which can solve the environment.
> >
> > It is worrying that this was not mentioned, in the same way that it is worrying that the POET parameters were not mentioned, see [4] from the previous comment.

---

> > > ### Author Response · Authors · 2021-11-30
> > > **Details of tuning POET in our experiment**
> > >
> > > ```
> > > Given this large array of parameters, and their interactions, it is difficult to evaluate the extent to which POET was well-tuned. ‘’, ‘’Could you please point me to details about how this was done? I can only find mention of the re-scaling factor for the reward (in appendix B), but there are many more parameter of POET.
> > > ```
> > >
> > > - **For most hyperparameters, we follow the same setting in the original code of POET** (https://github.com/uber-research/poet). We only tuned the hyperparameters related to the task/environment properties in our experiments by grid search because they are different from the tasks/ environments used in the POET paper. We list the tuned hyperparameters in the table below:
> > >
> > > | Hyperparameter | Tuning set | Final picked value |
> > > |:--|:--|:--|
> > > | Max_num_envs | {20, 30, 40} | 40 |
> > > | Max_children | {6, 8 ,10} | 8 |
> > > | Reproduction threshould| {70, 80, 90, 100} | 80|
> > > | Nearest neighbor k | [4, 8] | 6 |
> > > | Lower criteria | {5, 10, 15, 20, 25} | 10 |
> > > | Higher cirteria | {125, 130, 135, 140, 145, 150} | 130 |
> > > | Noise_devi| [0.05, 0.15] | 0.1 |
> > > | $N_{transfer}$ | {25, 30} | 25 |
> > >
> > > - We understand your worries about the importance/sensitivity of hyperparameter tuning to the performance of curriculum RL methods that are built upon some human-designed criteria, e.g., POET. We agree that this is a limitation of conventional curriculum methods. **However, we aim to remove such a limitation in developing EAT-C** by end-to-end learning (a planning policy and an environment generator) to “automatically” generate a curriculum of sub-tasks and associated environments adaptive to the progress of RL.
> > >
> > > - Please let us know if the above addresses all your concerns or not. Thanks!

---

> > > > ### Comment · Reviewer_o38w · 2021-11-30
> > > > **Thank you for the details of tuning POET in the experiments**
> > > >
> > > > Thank you for the response.  It is certainly clarifying to have the details on how the hyperparameters were tuned.
> > > >
> > > > >We only tuned the hyperparameters related to the task/environment properties in our experiments by grid search because they are different from the tasks/ environments used in the POET paper.
> > > >
> > > > Gird search over this space certainly seems adequate.  Even if the best parameters are not in the space, I would see that as a fault of POET for being too hard to transfer to new environments.  I appreciate that your method appears to be more resilient in this regard.  It would be important to include this information in the appendix, as it would allow future researchers to recreate the results, but otherwise this seems to address my concerns.
> > > >
> > > > Just to be clear you ran the ~20,000 combinations of these parameters for about 1 million environment training steps and took the best performing set?

---

> > > > > ### Author Response · Authors · 2021-11-30
> > > > > **Response to Reviewer o38w’s remaining concerns**
> > > > >
> > > > > We are glad to hear that several main concerns raised by you have been addressed by our reply and new experiments! Here is our reply to your remaining concerns:
> > > > >
> > > > > - A complete grid search over all the listed hyperparameters is too costly. Hence, we partitioned these hyperparameters into four groups of strongly related ones (listed below) and applied grid search within each group. For Group 2-4, we started from the default values used in the original POET paper. Since our environments are different, we cannot start from the default hyperparameters for Group 1 used in POET paper. Therefore, we tuned the hyperparameters from Group 1 to Group 4 in a greedy manner. We also tried different orders for Group 2-4. This resulted in hundreds of combinations of hyperparameters and we ran 0.8 millions environment training steps for each combination. We then chose the best combination with the highest success rate for all experiments.
> > > > >
> > > > > - The partition of hyperparameters in POET:
> > > > >
> > > > >    Group 1: The criteria for mutating environments: {lower criteria, higher criteria, reproduction_threshould}
> > > > >
> > > > >    Group 2: Pool size of environments/ mutation environments: {max_num_envs, max_children}
> > > > >
> > > > >    Group 3: Exploration and exploitation: {$N_{transfer}$,  noise_devi}
> > > > >
> > > > >    Group 4: Novelty bonus of mutating: {nearest_neighbor_k}
> > > > >
> > > > > Please let us know if you still have any remaining concerns. Thanks!

---

> > ### Comment · Reviewer_o38w · 2021-11-27
> > **Question about Q5**
> >
> > >Q5: ‘’the environment adversary could make unsolvable environments’’
> > We always conduct a connectivity test to the generated environments so the accepted environments used for training are all solvable. We have the same test for other baselines. It is worth noting that the environment generator in EAT-C is always trained on solvable environments so it tends to generate solvable ones. This is verified in our experiments: we observed that most of its generated environments can pass the connectivity test.
> >
> > This appears to be a critical aspect of the experiments which I do not see in the paper.  This is especially concerning as it could be vital to mitigating the failure mode of generating many unsolvable environments, so running an ablation to see if this is critical to the method seems important.  It also seems like the requirement that you can run a connectivity test on the training environment is a strong limitation of the applicability of the approach, that should be mentioned more prominently.
> >
> > >It is worth noting that the environment generator in EAT-C is always trained on solvable environments so it tends to generate solvable ones.
> >
> > To be clear, is this because the environments are pre-filtered by the connectivity test?  If that is the case, then EAT-C would generate unsolvable levels without this pre-filtering step making it critical to the success of the algorithm.

---

> > > ### Author Response · Authors · 2021-11-30
> > > **Response to questions of Q5**
> > >
> > > ```
> > > It also seems like the requirement that you can run a connectivity test on the training environment is a strong limitation of the applicability of the approach, that should be mentioned more prominently.
> > > ```
> > > - **Curriculum RL by its setting can control and determine the environments to train the RL agent,** because a curriculum is defined to select or generate the tasks and environments to train an RL agent. Since it has control of the environment, it has the capability to check the connectivity and select only solvable environments for learning. Curriculum RL can be conducted in a simulator so it is easy to let the curriculum have the control. **Reviewer Zhvq also argues that a curriculum RL method should be able to control everything.**
> > > - **The connectivity test is common in navigation as well as many complicated and realistic environments/tasks [1][2].** For example, in maze tasks like 2D-pusher, modifying the obstacles easily results in unsolvable environments. **Whether to apply the test is a property of an environment/task instead of a limitation of our method only**: to avoid wasting computation on unsolvable ones, most methods adopt the connectivity test by default when applied to such an environment/task. For some simpler environments/tasks, e.g., BipedalWalker mainly used in the original ALP-GMM and POET paper, connectivity test is not required by any method. However, as we show in the table below, **ALP-GMM and POET generate more unsolvable environments than EAT-C when applied to 2D-pusher.**
> > > - **Removing the connectivity test will not heavily change the final training results** because the agent gets no effective reward from unsolvable environments. However, **removing the test does affect the efficiency** because the agent has to waste time on unsolvable environments. As the new experiments we will show later, **EAT-C generates fewer unsolvable environments than other baselines**, so removing the test will not change the advantage of EAT-C on training efficiency.
> > > - Due to the nature of the environments in this paper, we applied the connectivity test to every method evaluated, so **the comparison is fair to all methods.** To evaluate how these methods’ efficiency is affected by the removal of the connectivity test, **in the table below, we report the percentage of unsolvable environments generated/sampled by different methods** at different stages of the training. It shows that (1) the unsolvable environments generated by all the three methods drastically decrease to <2% after 10 millions steps so they only affect the efficiency of early training stages; (2) EAT-C generates much less unsolvable environments than other baselines so it is still more efficient when the connectivity test is removed.
> > >
> > > |Env. steps | 2.0 million | 8.0 million | 10.0 million |
> > > |:--|:--|:--|:--|
> > > | EAT-C | 6.046%  |2.015% | 1.131% |
> > > | POET| 10.889% | 2.510% | 1.586% |
> > > | ALP-GMM |13.74% |  3.490% |1.697% |
> > >
> > >
> > > [1] Anthony Francis, et al., Long-Range Indoor Navigation with PRM-RL, 2020;
> > >
> > > [2] Yingjun Pei, et al., Learning Representations in Reinforcement Learning: an Information Bottleneck

---

> > > > ### Comment · Reviewer_o38w · 2021-11-30
> > > > **Response to response to questions of Q5**
> > > >
> > > > I appreciate the author's clarifications on this point.
> > > >
> > > > >Since it has control of the environment, it has the capability to check the connectivity and select only solvable environments for learning.
> > > >
> > > > There is a distinction here between control of the environment and knowing if the environment is solvable, the second is often strictly stronger.  For instance, if you were trying to train an agent to solve a bin-packing problems, you could have control over all of the bins and the packages but knowing if it was solvable would be NP-Hard.  Moreover, if we required an already effective bin-packing solver to train an RL agent then it would limit the impact of solving the problem with RL.
> > > >
> > > > > Removing the connectivity test will not heavily change the final training results
> > > >
> > > > It is not clear to me that the results would remain the same, but given that it is used for all the baselines I agree that the evaluation is fair, and I do not believe that it is a limitation that should prevent acceptance.  I am only concerned about it being mentioned as a critical part of the algorithm and a potential limitation.

---

> ### Author Response · Authors · 2021-11-23
> **Response to reviewer o38w (PART 1)**
>
> Thank you for your time and suggestions! We address your main concern as below:
>
> ###  Q1: ‘’it is not clear if what is achieved by this approach could be achieved by running existing HRL techniques in the context of adversarial environment design.’’
> - **HRL + adversarial environment design cannot perform well (it can be even worse than HRL) without the two novel strategies proposed in EAT-C**: (1) the mutual training between planner and RL agent, and the learnable environment generator (EG); and (2) the easy-to-hard sub-task curriculum for every model based on the sub-task tree.
> - First, **HRL can be inefficient without a correct curriculum**, as demonstrated in our experiments. Here is the reason: HRL trains the high-level and low-level policies in a top-down manner, i.e., **learning hard tasks first before easy ones**. This is inefficient due to the sparse rewards on the hard tasks at the beginning. On the contrary, we train the RL agent in a bottom-up manner so it learns to complete easier sub-tasks with dense rewards at first before addressing the harder ones. This easy-to-hard curriculum significantly improves the efficiency of RL.
> - Due to the same reason above, **combining HRL with adversarial environment design will make the efficiency even poorer** because the more challenging environments result in more sparse rewards at the beginning of training. Hence, such a strategy might easily fail and be stuck at the very beginning.
> - Secondly, **HRL does not have a mutual training scheme** between the high-level and low-level policies as the one between the planner and RL agent in EAT-C. Mutual training is very important to generate high-quality sub-tasks matching the need of the RL agent’s learning progress. However, the high-level policy in HRL is solely trained by the sparse environment reward, which is less efficient.
> - Thirdly, directly designing an adversarial environment for each sub-task is infeasible or non-trivial since the sequential decision-making of RL is often nondifferentiable. It is also costly because it needs to roll out the RL policy for each sub-task at first. Instead, **we train an environment generator (EG) in EAT-C in an end-to-end manner** so it can automatically produce the adversarial environment for each sub-task without any rollout or back-prop through trajectory.
> - In addition, we cannot efficiently train the EG model in HRL without our proposed bottom-up curriculum because training EG uses the RL agent’s reward, which is sparse if using the top-down hard-to-easy training strategy in HRL.
>
> ### Q2: ‘’the ablation study in Figure2 (b) leads me to believe that the adversarial environment design component is relatively unimportant,’’
>
> - **Figure 2(b) CANNOT reflect the main advantages brought by the adversarial environment generator**, i.e., the generalization and robustness of the RL agent when deployed to new or perturbed environments, **because Figure 2(b) evaluates all methods on the same environment used for training** (but on different tasks) during the test phase. This was mentioned in the 1st paragraph of Section 5.2.
> - To evaluate the generalization and robustness, which are the advantages due to the adversarial environment generator, we add a new ablation study that evaluates different methods on **multiple new random environments during the test phase**. The new results are reported in the table below, which shows **a large gap (80.24 vs. 42.23) between EAT-C and EAT-C (remove EG)**. This demonstrates that EG is important to improving the generalization and robustness of the RL policy.
>
> |Test setting| Multiple New Random Environments | Training Environment |
> |:--|:--|:--|
> | EAT-C | 80.24$\pm$12.25 | 92.04$\pm$6.49 |
> | EAT-C (remove EG) | 42.23$\pm$10.34 | 85.47$\pm$9.12 |
> | EAT-C (remove planner) | 27.58 $\pm$ 14.67 | 46.02$\pm$10.37 |
> | SAC | 20.83 $\pm$ 7.24 | 39.62 $\pm$ 12.25|
> | Hierarchical RL | 22.04 $\pm$ 10.44| 68.27 $\pm$ 6.99|

---

> > ### Comment · Reviewer_o38w · 2021-11-27
> > **Response to (PART 1)**
> >
> > > **HRL + adversarial environment design cannot perform well (it can be even worse than HRL) without the two novel strategies proposed in EAT-C**: (1) the mutual training between planner and RL agent, and the learnable environment generator (EG); and (2) the easy-to-hard sub-task curriculum for every model based on the sub-task tree.
> > >
> >
> > I think this is a central claim of the paper, but I do not see empirical evidence for this. If this is right then the natural baseline of using HRL in an environment designed by POET or by being built directly by an adversary would both fail dramatically.  That would be a strong empirical argument justifying the added complexity, and we would not need the following intuitive arguments.  The arguments you present below could easily be correct or incorrect and largely rely on intuition about how these two pretty complicated algorithms would interact.  Our community has a long history of incorrect intuitive arguments in settings much less complex than this.  I believe the empirical evaluation is necessary to tell if any of these intuitive arguments are correct.  Below I take issue with a few of these intuitive arguments directly, but I think it would be difficult to come to a resolution on these points without experiment.
> >
> > > **learning hard tasks first before easy ones**
> >
> > This seems like what the environment design/curricula would be aiming to fix?   It could be that POET would provide a good curriculum to HRL on its own.
> >
> > > **combining HRL with adversarial environment design will make the efficiency even poorer** because the more challenging environments result
> >
> > Most environment design methods are aimed at being curricula, so, if they work as designed, they should make the problem easier not harder.
> >
> > > Thirdly, directly designing an adversarial environment for each sub-task is infeasible or non-trivial since the sequential decision-making of RL is often non-differentiable. It is also costly because it needs to roll out the RL policy for each sub-task at first. Instead, **we train an environment generator (EG) in EAT-C in an end-to-end manner** so it can automatically produce the adversarial environment for each sub-task without any rollout or back-prop through trajectory.
> >
> > These challenges are the same as those faced by environment design broadly, those methods also learn systems to generate adversarial environments without needing rollouts.  The difference in this case is that you generate different environments for each subtask and do not present empirical evidence that this added complexity is necessary or worth the effort.
> >
> >
> > > **Q2: ‘’the ablation study in Figure2 (b) leads me to believe that the adversarial environment design component is relatively unimportant,’’**
> > >
> >
> > Thanks for the clarificaition.  I was incorrect on this point, and the new results mak it clear the adversarial environment design is important at least for robustness.

---

> > > ### Author Response · Authors · 2021-11-30
> > > **We add the new experiments you requested! Glad to see we have addressed some concerns!**
> > >
> > > Thank you for your response! We are glad to hear that some of your concerns have been successfully addressed by our reply. We understand your further concerns about the connectivity test, hyper-parameter tuning, and the adversarial environment generator. We provide more details and new experimental results below to specifically address your worries and support our previous claims. Here is our detailed reply:
> > >
> > > ```
> > > I think this is a central claim of the paper ... dramatically.
> > > ```
> > > - **Our current ablation study (Table 4 and Fig. 2(b)) already provided empirical evidence indicating that Hierarchical RL (HRL) easily fails under adversarial environment design. For example, the last row of Table 4** shows that the success rate of a well trained HRL policy significantly degrades from **68.27 to 22.04** when changing from a fixed  environment to random perturbed environments. Hence, HRL will perform even worse and easily fail with no effective reward if changing the random perturbed environments to adversarial ones.
> > > Without an appropriate curriculum such as the easier subtasks in EAT-C, HRL starts from learning hard and long-horizon tasks as RL. There exist plenty of works [1][2][3] showing that even **a very small adversarial perturbation to the environments can make RL fail or substantially inefficient.**
> > >
> > > ```
> > > This seems like ... aiming to fix? + Most environment design methods ... harder
> > > ```
> > > - Some environment design methods might be able to produce a curriculum of environments for HRL but **this cannot happen without sufficient random exploration over an enormous amount of possible environments. Such exploration is often highly inefficient and expensive** because evaluating every environment requires rollouts of the HRL policy. In contrast, EAT-C does not aim at producing a curriculum of environments but instead a curriculum of sub-tasks by more efficient sub-goal tree planning. The environment generator only needs to output one-time local modification to the environment in each subtask.
> > > - **Making tasks overly easier is NOT always helpful to improve RL if they are too trivial for the RL agent** so nothing can be learned from completing them, e.g., removing all obstacles from a maze. In EAT-C, the planner decomposes a hard task into a sequence of easy sub-tasks but some of them might be too similar (e.g., due to some symmetry of the environment) or too easy and thus less information to RL. Therefore, an adversarial modification to the environment of each sub-task is necessary to make these subtasks diverse and sufficiently challenging in order to provide more informative feedback to improve the RL policy.
> > >
> > > ```
> > > The difference in this case ... worth the effort.
> > > ```
> > > - Training an environment generator (EG) to locally modify environments for a curriculum of sub-task **is easier and results in more efficient learning** because (1) EG does not need to create a curriculum or fully explore the whole environment space, which is highly expensive and challenging in other environment design methods; (2) generating does NOT add complexity compared to other environment design methods. On the opposite, it REDUCES the complexity because easier sub-tasks can provide denser and more informative feedback than the original long-horizon task.
> > >
> > > - **In the following new experiments, as you suggested, we empirically demonstrate the advantage of our generated environments with the ones generated by POET**. Specifically, we replace EG in EAT-C with POET so both the path-planner and the RL agent are trained within the environments generated/mutated by POET. This is “(EG &rarr POET)” in the table below, which can be seen as an example of “HRL+POET”.
> > >
> > > |Env. steps | 2.5 million steps | 5.0 million steps | 7.5 million steps | 10.0 million steps |
> > > |:--|:--|:--|:--|:--|
> > > | EAT-C (original) | 22.07 $\pm$ 10.55 |39.76$\pm$ 13.46  |61. 24$\pm$  16.13 |69.45  $\pm$ 17.42 |
> > > | EAT-C (EG --> POET) |  16.44 $\pm$8.21 | 37.18 $\pm$14.25 | 49.94 $\pm$12.21 | 57.64 $\pm$10.93 |
> > > | POET | 20.32 $\pm$ 11.35 | 39.22 $\pm$ 10.79 | 43.10 $\pm$ 14.54 | 50.23 $\pm$ 12.83 |
> > >
> > > - The results show that applying some curriculum generated by existing methods such as **POET to HRL (i.e., EAT-C (EG --> POET)) can finally outperform POET but it is less efficient than POET** in early stages (before 5.0 million steps), because of the expensive and inefficient exploration of the environment space discussed above. On the other hand, **our original EAT-C significantly outperforms both POET and EAT-C (EG --> POET) on learning efficiency and final performance.** Hence, our sub-task curriculum with adversarial environments is more efficient than some existing curricula applied to HRL.
> > >
> > > [1] Lerrel Pinto, et al., Supervision via competition: Robot adversaries for learning tasks.
> > >
> > > [2] Rawal Khirodkar, et al.. Vadra: Visual adversarial domain randomization and augmentation.
> > >
> > > [3] Eugene Vinitsky, et al.. D-malt: Diverse multi-adversarial learning for transfer.

---

> > > > ### Comment · Reviewer_o38w · 2021-11-30
> > > > **Thank you for running this valuable experiment.**
> > > >
> > > > > “(EG &rarr POET)” in the table below, which can be seen as an example of “HRL+POET”.
> > > >
> > > > Thank you for running these experiments.  I appreciate the effort and they are certainly clarifying in showing how your method compares to the HRL + POET baseline.

---

### Official Review · Reviewer_7T33 · 2021-11-08

**Correctness:** 4
**Technical Novelty And Significance:** 3
**Empirical Novelty And Significance:** 3
**Recommendation:** 6
**Confidence:** 3

**Main Review:**

### Pros:

1. The authors address an important problem: a curriculum learning of challenging tasks under the sparse reward setting where current RL techniques are known to be inefficient. To me, the problem is real and practical.
2. The paper includes comprehensive experiments, including both qualitative analysis and quantitative results, to show the effectiveness of the proposed method. Thorough ablation studies are also performed.

### Cons:

1. Although the results look good on the two environments provided by the authors, providing additional experiments using more challenging benchmarks, such as 3D continuous control tasks, would strengthen the paper. For example, the 7DoF robotic arm problem from [1] seems quite relevant.
2. The think the writing can be improved. First, certain details are missing (e.g. how exactly the environment generation was implemented). Second, certain details (e.g. Algorithm pseudocode, model architecture, etc) can be moved to the appendix to make space for a Conclusion and Future work Section, which is noticeably missing. Third,, the figures 3 and 4 include texts and line names that is very small, and difficult to read (for someone who printed the paper using standard size). Please fix these.

[1] T. Jurgenson, Orly Avner, E. Groshev, and Aviv Tamar. Sub-goal trees - a framework for goal-based reinforcement learning. ArXiv, abs/2002.12361, 2020.

**Summary Of The Paper:**

This work introduces a method called Environment-Adversarial Sub-Task Curriculum  (EAT-C) wjocj automatically generates a curricuulm of task-environment couples for efficient RL. To this end, EAT-C trains two policies, in addition to the main decision-making policy. One policiy recursively decomposes hard task to coarse-to-fine subtasks, while the other performs adversarial modifications of the environment in each of the subtask. The authors show empirically that EAT-C leads to efficient training of the policy and superior generalization capabilities compared to existing baselines.

**Summary Of The Review:**

The authors address an important problem and provide comprehensive experimental details, including qualitative analysis, quantitative results and thorough ablations. However, further experiments on more challenging domains would strengthen the paper. Also, the writing can be improved.

---

> ### Author Response · Authors · 2021-11-23
> **Response to reviewer 7T33**
>
> Thank you for your time and suggestions! As you suggested, we have conducted experiments on the 7DoF robotic arm and reported the results in Appendix D.3 of the updated paper. We address your main concerns as below:
>
> ### Q1: ‘’How exactly the environment generation was implemented.’’
> - **We have uploaded a new version of the paper that contains the details of environment generation in Appendix A**. For EAT-C in the 2D-pusher experiment, we represent the environment by $\theta$ that comprises multiple tuples in the form of $(x_i,y_i,w_i,h_i)_n$ , each defining a rectangle obstacle with location coordinates $(x_i, y_i)$ and size (width and height) $(w_i,h_i)$. The environment generator (EG) can modify each obstacle in the environment by changing its location and size, where the change is $b_t = (\Delta x_i, \Delta y_i, \Delta w_i, \Delta h_i)$, i.e., an action taken by EG. We also constrain the changes in $b_t$ from being too large by thresholding, similar to adversarial attacks.
>
> ### Q2: “make room for conclusion and future works”
> - We have added a conclusion section in our new version, i.e.,:
> We propose a mutual learning and auto-curriculum framework “EAT-C” to improve the efficiency of RL on long-horizon tasks as well as its generalization and robustness to new environments. EAT-C trains a planner to decompose a hard task into coarse-to-fine sequences of sub-tasks providing an easy-to-hard curriculum to train an RL agent, while an adversarial environment generator modifies these sub-tasks to be diverse and more informative to learn. The three policies are trained with data collected by each other. On two types of tasks, EAT-C outperforms a diverse set of baselines, e.g., curriculum-based RL, hierarchical RL, and planning-based methods. It has the potential to be applied to more complicated tasks with dynamic environments or visual inputs such as games, which will be covered in our future works.
>
>
> ### Q3: “the figures 3 and 4 include texts and line names that is very small”
> - We have added larger versions of these two figures in the Appendix and we will try to make room for them in the main paper later.

---

### Author Response · Authors · 2021-11-23
**Summary of changes in Rebuttal Version**

We appreciate all reviewers for their constructive suggestions! We carefully addressed all the raised concerns and accordingly modified the paper. We highlighted the modified parts with red color in the new version and here is a summary of the new changes for your convenience:
- [**New experiments**] We added **new experiments of a more complicated task on a 7DoF robotic arm** and the results are reported in Appendix D.3, showing the advantage of EAT-C over other baselines on different types of tasks.
- [**New experiments**] We added a group of **new ablation study experiments** in Appendix D.2. It aims at evaluating and comparing the performance of different methods **on multiple random and unseen environments during the test phase**. The previous ablation study evaluates them in the fixed training environment.
- We added details of the implementation of the environment parameterization and environment generator (EG) in Appendix A (A.1 - the details of EAT-C for 2D-pusher; A.2 - the details of EAT-C for the discrete tasks; A.3 - Baseline methods)
- We summarized the discussion of the suggested reference in Appendix E.
- We added a conclusion section.
- We added a discussion about all the suggested references in Section 2.
- We improved the statements on Page 2.
- We moved the pseudocode of Algorithm 1 & 2 to Appendix C.
- We added larger versions of Figure 3 & 4 in Appendix D.1.
- We modified the caption of Figure 2.
- We fixed the typo on Page 3.

---

### Decision · Program_Chairs · 2022-01-20

**Decision:**

Reject

**Comment:**

I thank the authors for their submission and active participation in the discussions. This papers is borderline with three reviewers leaning towards acceptance [3c96,7T33,Zhvq] and one leaning towards rejection [o38w]. Reviewer o38w's main concerns are around the lack of details about how the baselines were tuned and missing training details (specifically the connectivity test used to reject candidate environments). During discussion both, reviewers Zhvq and 7T33, agree that the paper requires substantial restructuring/rewriting to properly address the reviewer's feedback which is currently mostly addressed in the appendix. Based on the discussion with reviewers, my assessment is that this paper is not ready for publication at this point and that it will benefit greatly from another iteration. I want to very strongly encourage the authors to further improve their paper based on the reviewer feedback.